# Transformer Meets Twicing: Harnessing Unattended Residual Information

**Laziz U. Abdullaev**
Department of Mathematics
National University of Singapore
`laziz.abdullaev@u.nus.edu`

**Tan M. Nguyen**
Department of Mathematics
National University of Singapore
`tanmn@nus.edu.sg`

## ABSTRACT

Transformer-based deep learning models have achieved state-of-the-art performance across numerous language and vision tasks. While the self-attention mechanism, a core component of transformers, has proven capable of handling complex data patterns, it has been observed that the representational capacity of the attention matrix degrades significantly across transformer layers, thereby hurting its overall performance. In this work, we leverage the connection between self-attention computations and low-pass non-local means (NLM) smoothing filters and propose the Twicing Attention, a novel attention mechanism that uses *kernel twicing procedure* in nonparametric regression to alleviate the low-pass behavior of associated NLM smoothing with compelling theoretical guarantees and enhanced adversarial robustness. This approach enables the extraction and reuse of meaningful information retained in the residuals following the imperfect smoothing operation at each layer. Our proposed method offers two key advantages over standard self-attention: 1) a provably slower decay of representational capacity and 2) improved robustness and accuracy across various data modalities and tasks. We empirically demonstrate the performance gains of our model over baseline transformers on multiple tasks and benchmarks, including image classification and language modeling, on both clean and corrupted data. The code is publicly available at `https://github.com/lazizcodes/twicing_attention`.

## 1 INTRODUCTION

Attention mechanisms and transformers (Vaswani et al., 2017) have achieved state of the art performance across a wide variety of tasks in machine learning (Khan et al., 2022; Lin et al., 2022; Tay et al., 2022) and, in particular, within natural language processing (Al-Rfou et al., 2019; Baevski & Auli, 2018; Dehghani et al., 2018; Raffel et al., 2020; Dai et al., 2019), computer vision (Liu et al., 2021; Touvron et al., 2021; Radford et al., 2021), and reinforcement learning (Janner et al., 2021; Chen et al., 2021). They have also demonstrated strong performance in knowledge transfer from pretraining tasks to various downstream tasks with weak or no supervision (Radford et al., 2018; 2019; Devlin et al., 2018). At the core of these models is the dot-product self-attention mechanism, which learns self-alignment between tokens in an input sequence by estimating the relative importance of each token with respect to all others. The mechanism then transforms each token into a weighted average of the feature representations of the other tokens with weights proportional to the learned importance scores. The relative importance scores capture contextual information among tokens and are key to the success of the transformer architecture (Vig & Belinkov, 2019; Tenney et al., 2019; Cho et al., 2014; Tran et al., 2025; Parikh et al., 2016; Lin et al., 2017; Nguyen et al., 2021).

Even though deep transformer-based models have achieved notable success, they are prone to the representation collapse issue, where all token representations become nearly identical as more layers are added. This phenomenon, often referred to as the "over-smoothing" problem, substantially reduces the transformers' ability to represent diverse features (Shi et al., 2022; Wang et al., 2022; Nguyen et al., 2023a; Devlin et al., 2018; Nguyen et al., 2024a).

Correspondence to: `laziz.abdullaev@u.nus.edu`

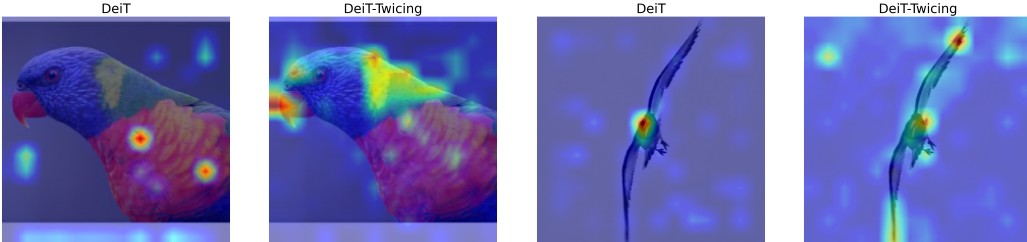

Figure 2: DeiT (Touvron et al., 2021) and DeiT-Twicing (ours) attention heatmaps. Our model shows better representational capacity compared to the baseline by paying attention to more meaningful parts of objects while DeiT attention scores are collapsed to one or few points.

To demonstrate this phenomenon, we analyze the average cosine similarity between token pairs across layers in a softmax transformer trained for the Imagenet classification tasks. As shown in Figure 1, the cosine similarity between tokens increases with depth. In the final layers, the cosine similarity scores are just under 0.9, suggesting a high level of similarity among token representations.

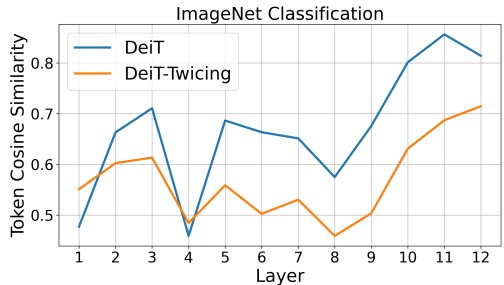

Figure 1: Average token cosine similarities across layers of DeiT and DeiT-Twicing over 100 random samples. Our model retains better token diversity compared to the baseline.

A prior line of research explores representation collapse in transformers through the lens of image denoising, showing that self-attention computation is equivalent to a gradient descent step towards minimizing energy functional that promotes smoothness in the input image (Nguyen et al., 2023b; Gilboa & Osher, 2007). Additionally, investigating the over-smoothing phenomenon from a graph-based perspective has gained significant attention in recent studies (Wu et al., 2023; Shi et al., 2022; Nguyen et al., 2023a; 2024b).

**Contribution.** In this work, we take the connection between the self-attention mechanism and the nonlocal-means image smoothing filter (Buades et al., 2005) further, and show that rapidly vanishing eigenvalues of associated NLM filter across iterations is a major cause of representation collapse in transformers. The NLM similarity matrix, the heart of NLM smoothing procedure, computes pairwise similarities between image patches based on intensity differences, effectively serving as a weight matrix in the smoothing process. We then propose the Twicing Attention, a novel attention mechanism, redesigned from the modified NLM smoothing operation that is tailored to decrease the rate of decay of the eigenvalues of the NLM similarity matrix and thereby offering advantages over the standard NLM based self-attention. In particular, we establish a connection between our modification technique and the twicing kernels in nonparametric regression (Stuetzle & Mittal, 1979; Newey et al., 2004; Abdous, 1995), uncovering the modified NLM filter's ability to exploit meaningful information in the residuals of each transformer layer after applying a smoothing operation. In summary, our contributions are three-fold:

1. We develop the novel Twicing Attention mechanism, a self-attention mechanism variant that promotes better token diversity across transformer layers which also enjoys enhanced robustness.

2. We develop a theoretical framework highlighting the effectiveness of Twicing Attention in mitigating representational collapse by decelerating the rate of eigenvalue vanishing phenomenon.

3. We show, through the lens of twicing kernels in nonparametric regression, how unattended but useful residual information between self-attention input and output can be used as a self-correction at each transformer layer.

Moreover, we empirically validate the performance improvements of Twicing Attention over standard self-attention in large-scale tasks such as ImageNet-1K classification (Touvron et al., 2021),

ADE20K image segmentation (Strudel et al., 2021) and WikiText-103 language modelling (Merity et al., 2016), and offer additional insights into its implementation with minimal additional computational overhead. We also assess its robustness against adversarial attacks, data contamination, and various distribution shifts.

**Organization.** The paper is written in the following structure: In Section 2, we introduce the reader with some background context on self-attention mechanism and its connection to image smoothing operation as a warm up to achieve better readability overall. In Section 3, we leverage the connection between self-attention mechanism and nonlocal-means (NLM) smoothing filters to show that representation collapse phenomenon is particularly caused by low-pass behaviour of such filtering procedure. Then, we propose a novel technique to alleviate the low-pass behaviour of associated NLM smoothing, thereby enabling a redesign of the standard self-attention mechanism with better expressive power across the transformer layers. In Section 4, we present our experimental results using Twicing Attention while Section 6 contains a brief overview of related work in the literature. Finally, we end with concluding remarks in Section 7 and defer most of the technical proofs and derivations as well as extra experimental observations to appendix.

## 2 BACKGROUND

### 2.1 SELF-ATTENTION MECHANISM

Given an input sequence $\mathbf{X} = [\boldsymbol{x}_1, \dots, \boldsymbol{x}_N]^\top \in \mathbb{R}^{N \times D_x}$ of $N$ feature vectors, the self-attention mechanism transforms the input to $\mathbf{U} := [\boldsymbol{u}_1, \dots, \boldsymbol{u}_N]^\top \in \mathbb{R}^{N \times D_x}$ as follows:

$$\boldsymbol{u}(i) = \sum_{j=1}^N \text{softmax}\left(\frac{\boldsymbol{x}_i^\top \mathbf{W}_K^\top \mathbf{W}_Q \boldsymbol{x}_j}{\sqrt{D}}\right) \mathbf{W}_V \boldsymbol{x}_j$$

$$= \sum_{j=1}^N \text{softmax}\left(\frac{\boldsymbol{q}_i^\top \boldsymbol{k}_j}{\sqrt{D}}\right) \boldsymbol{v}_j \tag{1}$$

for $i = 1, \dots, N$, where $\text{softmax}(a_j) := \text{softmax}(\boldsymbol{a})_j$ for $\boldsymbol{a} = [a_1, \dots, a_N]$ is an abuse of notation for convenience. The vectors $\boldsymbol{q}_i, \boldsymbol{k}_j$, and $\boldsymbol{v}_j$, $j = 1, \dots, N$, are the query, key, and value vectors, respectively. They are computed as $\mathbf{Q} := [\boldsymbol{q}_1, \dots, \boldsymbol{q}_N]^\top = \mathbf{X}\mathbf{W}_Q^\top \in \mathbb{R}^{N \times D}$, $\mathbf{K} := [\boldsymbol{k}_1, \dots, \boldsymbol{k}_N]^\top = \mathbf{X}\mathbf{W}_K^\top \in \mathbb{R}^{N \times D}$, and $\mathbf{V} := [\boldsymbol{v}_1, \dots, \boldsymbol{v}_N]^\top = \mathbf{X}\mathbf{W}_V^\top \in \mathbb{R}^{N \times D_v}$, where $\mathbf{W}_Q, \mathbf{W}_K \in \mathbb{R}^{D \times D_x}, \mathbf{W}_V \in \mathbb{R}^{D_v \times D_x}$ are the weight matrices. Eqn. 1 can be expressed in matrix form as:

$$\mathbf{U} = \text{softmax}\left(\frac{\mathbf{Q}\mathbf{K}^\top}{\sqrt{D}}\right)\mathbf{V}, \tag{2}$$

where the softmax function is applied row-wise to the matrix $\mathbf{Q}\mathbf{K}^\top/\sqrt{D}$. We refer to transformers built with Eqn. 2 as standard transformers or just transformers.

### 2.2 NONLOCAL VARIATIONAL DENOISING FRAMEWORK FOR SELF-ATTENTION

Based on the framework established by (Nguyen et al., 2023b), we first consider the output matrix $\mathbf{U} := [\boldsymbol{u}(1), \cdots, \boldsymbol{u}(N)]^\top \in \mathbb{R}^{N \times D}$ in self-attention as given by Eqn. 2 in Section 2.1. Let $\Omega \subset \mathbb{R}, x \in \Omega$, and $\boldsymbol{u}(x) := [u_1(x), \cdots, u_D(x)]^\top$ be a real vector-valued function, $\boldsymbol{u} : \Omega \to \mathbb{R}^D, \boldsymbol{u} \in L^2(\Omega)$. The output matrix $\mathbf{U}$ in self-attention discretizes the function $\boldsymbol{u}(x)$ with respect to $x$. In the context of signal/image denoising, $\mathbf{U}$ can be considered as the *desired clean signal*, and $\boldsymbol{u}(x)$ is its corresponding intensity function denoting the signal values at the position $x \in \Omega$. We further let the observed intensity function $\boldsymbol{f}(x)$ denote the values of the *observed noisy signal* at $x \in \Omega, \boldsymbol{f} : \Omega \to \mathbb{R}^D, \boldsymbol{f} \in L^2(\Omega)$. For example, $\boldsymbol{f}(x)$ can be given as

$$\boldsymbol{f}(x) = \boldsymbol{u}(x) + \boldsymbol{n}(x), \tag{3}$$

where $\boldsymbol{n}$ is the additive noise (see Eqn. 1 of (Buades et al., 2005)). We wish to reconstruct $\boldsymbol{u}(x)$ from $\boldsymbol{f}(x)$. Following the variational denoising method proposed in (Gilboa & Osher, 2007), the denoised image $\boldsymbol{u}(x)$ can be obtained by minimizing the following regularized functional with respect to $\boldsymbol{u}$:

$$E(\boldsymbol{u}, \boldsymbol{f}) = J_w(\boldsymbol{u}) + G(\boldsymbol{u}, \boldsymbol{f}) = \frac{1}{2}\int_{\Omega \times \Omega} \|\boldsymbol{u}(x) - \boldsymbol{u}(y)\|_2^2 w(x, y) dx dy + \frac{\lambda}{2}\int_\Omega \|\boldsymbol{u}(x) - \boldsymbol{f}(x)\|_2^2 dx. \tag{4}$$

Here, $J_w(\boldsymbol{u}) = \frac{1}{2} \int_{\Omega \times \Omega} \|\boldsymbol{u}(x) - \boldsymbol{u}(y)\|_2^2 w(x, y) dx dy$ is a nonlocal functional of weighted differences. The weights $w(x, y)$ represent the affinity between signal values at positions $x$ and $y$. For example, for images, $w(x, y)$ captures the proximity between pixels $x$ and $y$ in the image. $J(\boldsymbol{u})$ works as a regularizer. Minimizing $J(\boldsymbol{u})$ promotes the smoothness of $\boldsymbol{u}$ and penalizes high-frequency noise in the signal as discussed in the next section. Adding the convex fidelity term $G(\boldsymbol{u}, \boldsymbol{f}) = \frac{\lambda}{2} \int_{\Omega} \|\boldsymbol{u}(x) - \boldsymbol{f}(x)\|_2^2 dx$, with the regularization parameter $\lambda$, to the functional $J(\boldsymbol{u})$ allows the denoised signal $\boldsymbol{u}(x)$ to preserve relevant information in the observed noisy signal $\boldsymbol{f}(x)$. In the following section, we show that NLM algorithm for image filtering corresponds to a fixed point iteration step to solve the stationary point equation of $J_\omega$.

## 2.3 TRANSFORMERS IMPLEMENT ITERATIVE SMOOTHING

Note that the functional $J_w(\boldsymbol{u})$ imposes a stronger penalty on discontinuities or sharp transitions in the input signal $\boldsymbol{u}$, thereby promoting smoothness throughout the signal. To get the minimizer of $E(\boldsymbol{u}, \boldsymbol{f})$, we consider the following system of equation:

$$\frac{\partial E(\boldsymbol{u}(x), \boldsymbol{f}(x))}{\partial \boldsymbol{u}(x)} = \frac{\partial J_w(\boldsymbol{u}(x))}{\partial \boldsymbol{u}(x)} + \lambda(\boldsymbol{u}(x) - \boldsymbol{f}(x)) = 0, \quad \forall x \in \Omega. \tag{5}$$

Direct gradient calculation, as detailed in Appendix A.3, then yields

$$\int_{\Omega} (\boldsymbol{u}(x) - \boldsymbol{u}(y)) w(x, y) dy + \lambda(\boldsymbol{u}(x) - \boldsymbol{f}(x)) = 0, \quad \forall x \in \Omega. \tag{6}$$

Rearranging the terms in Eqn. 6, we obtain

$$\boldsymbol{u}(x) = \frac{\lambda \boldsymbol{f}(x) + \int_{\Omega} w(x, y) \boldsymbol{u}(y) dy}{\lambda + \int_{\Omega} w(x, y) dy}, \quad \forall x \in \Omega. \tag{7}$$

It is worth noting that Eqn. 7 becomes NLM filter with weights $w(x, y)$ when $\lambda = 0$ (see Eqn. 2 of (Buades et al., 2005)). In order to establish a connection between NLM and self-attention, let $\boldsymbol{k}(x) := [k_1(x), \ldots, k_D(x)]^\top$ be a real vector-valued function, $\boldsymbol{k} : \Omega \to \mathbb{R}^D, \boldsymbol{k} \in L^2(\Omega)$. Similar to $\boldsymbol{u}(x)$ and $\boldsymbol{v}(x)$, we can discretize $\boldsymbol{k}(x)$ on a 1-D grid to attain the key vectors $\boldsymbol{k}(1), \ldots, \boldsymbol{k}(N) \in \mathbb{R}^D$, which form the key matrix $\mathbf{K} := [\boldsymbol{k}(1), \ldots, \boldsymbol{k}(N)]^\top \in \mathbb{R}^{N \times D}$ in self-attention as defined in Eqn. 2. Neglecting the symmetry of the kernel, we choose $w(x, y) = \exp(\boldsymbol{q}(x)^\top \boldsymbol{k}(y)/\sqrt{D})$ and rewrite Eqn. (7) with $\lambda = 0$ as follows:

$$\boldsymbol{u}(x) = \frac{\int_{\Omega} \exp(\boldsymbol{q}(x)^\top \boldsymbol{k}(y)/\sqrt{D}) \boldsymbol{u}(y) dy}{\int_{\Omega} \exp(\boldsymbol{q}(x)^\top \boldsymbol{k}(y)/\sqrt{D}) dy}, \quad \forall x \in \Omega. \tag{8}$$

In line with the methodology proposed by (Nguyen et al., 2023b), the Monte-Carlo discretization of the above expression with respect to $x, y \in \Omega$ yields

$$\boldsymbol{u}(i) = \frac{\sum_{j=1}^{N} \exp(\boldsymbol{q}(i)^\top \boldsymbol{k}(j)/\sqrt{D}) \boldsymbol{u}(j)}{\sum_{j=1}^{N} \exp(\boldsymbol{q}(i)^\top \boldsymbol{k}(j)/\sqrt{D})}, \tag{9}$$

for which the following iterative solver is a natural choice:

$$\begin{cases} \boldsymbol{u}^{\ell+1}(i) &= \dfrac{\sum_{j=1}^{N} \exp(\boldsymbol{q}(i)^\top \boldsymbol{k}(j)/\sqrt{D}) \boldsymbol{u}^{\ell}(j)}{\sum_{j=1}^{N} \exp(\boldsymbol{q}(i)^\top \boldsymbol{k}(j)/\sqrt{D})}, &\forall \ell \in \mathbb{N}, \\ \boldsymbol{u}^0(i) &= \boldsymbol{f}(i), \end{cases}$$

where $\ell$ is an iteration step. It can be seen that setting $\lambda = 0$ and $\boldsymbol{u}^{\ell}(j) = \boldsymbol{v}^{\ell}(j)$, one iteration step becomes

$$\boldsymbol{u}^{\ell+1}(i) = \frac{\sum_{j=1}^{N} \exp(\boldsymbol{q}(i)^\top \boldsymbol{k}(j)/\sqrt{D}) \boldsymbol{v}^{\ell}(j)}{\sum_{j=1}^{N} \exp(\boldsymbol{q}(i)^\top \boldsymbol{k}(j)/\sqrt{D})} = \sum_{j=1}^{N} \text{softmax}\left(\frac{\boldsymbol{q}(i)^\top \boldsymbol{k}(j)}{\sqrt{D}}\right) \boldsymbol{v}^{\ell}(j), \tag{10}$$

which is equivalent to the self-attention computation given by Eqn. (1).

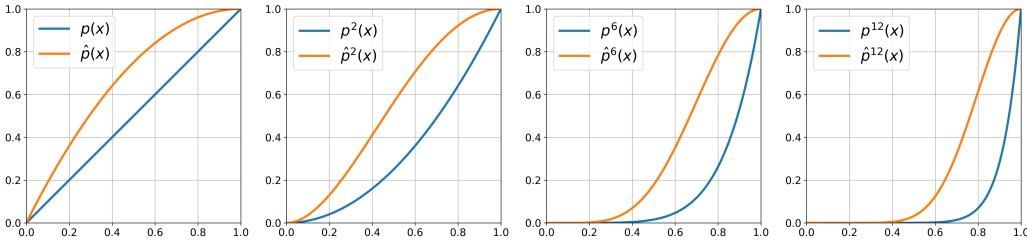

Figure 3: Dynamics of $p^n(x) = x^n$ and $\hat{p}^n(x) = (2x - x^2)^n$ for $n = 1, 2, 6, 12$.

## 3 HARNESSING UNATTENDED RESIDUAL INFORMATION VIA TWICING ATTENTION

In this section, we shall associate the representation collapse phenomenon with the rapidly vanishing spectrum of NLM similarity matrix during the iterative process, and then propose a method to alleviate this issue. Then, we will give deeper theoretical support for the modification method before proposing our Twicing Attention associated with this modified NLM filter.

### 3.1 VANISHING EIGENVALUES IN ITERATIVE NLM FILTERING

The denoising iteration can be written as the matrix-vector multiplication

$$\boldsymbol{u}^1 = \mathbf{D}^{-1}\mathbf{W}\boldsymbol{u}^0, \tag{11}$$

where $\mathbf{W}$ is an $N \times N$ matrix given by $\mathbf{W}_{ij} = w(i, j)$, and $\mathbf{D}$ is a diagonal matrix with $\mathbf{D}_{ii} = \sum_{j=1}^{N} \mathbf{W}_{ij}$. Introducing the averaging operator $\mathbf{A} = \mathbf{D}^{-1}\mathbf{W}$, the denoising iteration Eqn. 11 becomes $\boldsymbol{u}_d = \mathbf{A}\boldsymbol{u}$. The matrix $\mathbf{A}$ is conjugate to the positive definite matrix $\mathbf{S} = \mathbf{D}^{-1/2}\mathbf{W}\mathbf{D}^{-1/2}$ via $\mathbf{A} = \mathbf{D}^{-1/2}\mathbf{S}\mathbf{D}^{1/2}$. This implies that $\mathbf{A}$ has a complete set of right eigenvectors $\{\boldsymbol{\xi}_j\}_{j=1}^{N}$ and positive eigenvalues $1 = \lambda_1 \geq \lambda_2 \geq \cdots \geq \lambda_N > 0$. The largest eigenvalue is $\lambda_1 = 1$, corresponding to the trivial all-ones right eigenvector ($\mathbf{A}\mathbf{1} = \mathbf{1}$). We expand the signal vector $\boldsymbol{u}$ in the eigenbasis as $\boldsymbol{u} = \sum_{j=1}^{N} c_j \boldsymbol{\xi}_j$, where $c_j = \langle \boldsymbol{\xi}_j, \boldsymbol{u} \rangle$. Applying one step of NLM gives

$$\mathbf{A}\boldsymbol{u} = \sum_{j=1}^{N} c_j \mathbf{A}\boldsymbol{\xi}_j = \sum_{j=1}^{N} \lambda_j c_j \boldsymbol{\xi}_j.$$

Iteratively applying NLM $n$ times, however, yields

$$\mathbf{A}^n \boldsymbol{u} = \sum_{j=1}^{N} \lambda_j^n c_j \boldsymbol{\xi}_j \tag{12}$$

by the same argument. Eqn. 12 reveals that denoising is accomplished by projecting the image onto the basis $\{\boldsymbol{\xi}_j\}_{j=1}^{N}$ and attenuating the contributions of the eigenvectors associated with smaller eigenvalues. Observing that in Eqn. 12, the dynamics of eigenvalues are represented as

$$p^n(\mathbf{A})\boldsymbol{u} = \sum_{j=1}^{N} p^n(\lambda_j) c_j \boldsymbol{\xi}_j, \tag{13}$$

where $p(\lambda) = \lambda$, an identity polynomial whose iterations exhibit steep inclines near $\lambda = 1$ and declines sharply towards zero elsewhere. This results in the iterations converging rapidly toward a constant degenerate solution loosing salient information in the input.

### 3.2 LEVERAGING A QUADRATIC KERNEL TOWARDS BETTER INFORMATION CAPACITY

In this section, we revisit the eigenvector expansion of the matrix $\mathbf{A}$ as indicated in Eqn. 12. Although, in theory, high-frequency noise is effectively captured by the eigenvectors corresponding to the smallest eigenvalues, in practice, the iterative denoising process can also suppress the contributions of eigenvectors with larger eigenvalues, leading to potential information loss.

To address this issue, we work out an alternative polynomial dynamics $\hat{p}_n(\cdot)$, which aims to: (*eigenvalue enhancement*) maximally enhance eigenvalues such that $\hat{p}_n(\lambda) \geq p_n(\lambda)$ for all $\lambda \in [0,1]$, and (*0-1 boundedness*) ensure that values remain within the range $[0,1]$ to prevent any eigenvalue from exploding limits as $n \to \infty$, thereby ensuring $-\infty < 0 \leq \lim_{n\to\infty} \hat{p}_n(\lambda) \leq 1 < \infty$. Owing to the computational overhead associated with higher-degree polynomials, we limit our focus to quadratic polynomials. By setting $\hat{p}(0) = 0$, general form of such polynomials is given by $\hat{p}(\lambda) = a\lambda + b\lambda^2$. Conditions (*eigenvalue enhancement*) and (*0-1 boundedness*) imply $1 \geq \hat{p}(1) \geq p(1) = 1$, leading to $b = 1 - a$. This constraint reformulates $\hat{p}$ as:

$$\hat{p}(\lambda) = a\lambda + (1-a)\lambda^2. \tag{14}$$

Basic analysis reveals that $\hat{p}$ attains its maximum at $\lambda_a = \frac{a}{2(a-1)}$, which is feasible for all $a \notin (0,2)$. To satisfy condition (*0-1 boundedness*), we determine $a$ by solving $\hat{p}(\lambda_a) = 1$, yielding a unique solution of $a = 2$. This confirms that the optimal quadratic polynomial fulfilling both conditions (*eigenvalue enhancement*) and (*0-1 boundedness*) is $\hat{p}(\lambda) = 2\lambda - \lambda^2$. Figure 3 illustrates that $\hat{p}^n(\lambda)$ and $p^n(\lambda)$ perform similarly in discarding small eigenvalues near 0, essential for effective noise removal. However, $\hat{p}^n(\lambda)$ remains significantly larger near 1, thus better retaining the salient information captured by the input. This observation suggests using $2\mathbf{A} - \mathbf{A}^2$ as a candidate similarity matrix for smoothing the given input without drastically loosing mid-ranged eigenvalues and, thus, being capable of capturing more salient information.

## 3.3 WHY $2\mathbf{A} - \mathbf{A}^2$ HELPS: THEORETICAL GROUNDING

Now we shall attempt to provide deeper theoretical insights on the benefits of employing $2\mathbf{A} - \mathbf{A}^2$ as a step denoiser, or a similarity matrix in general. First, we demonstrate in Proposition 1 that it achieves substantially slower decay rate of representational capacity in the long run. The connection to twicing kernels, from which the paper title originates, is established and Proposition 2 is presented to demonstrate how these kernels effectively reduce estimation bias in nonparametric regression, another smoothing procedure associated with self-attention.

**Mitigating representation collapse.** To rigorously analyze the differences in denoising dynamics between the kernels $p(\mathbf{A}) = \mathbf{A}$ and $\hat{p}(\mathbf{A}) = 2\mathbf{A} - \mathbf{A}^2$, we define eigencapacity, which correlates with the model's information representation capacity, in Definition 1. Then, we demonstrate in Proposition 1 that the eigencapacity of the former kernel decays at a significantly faster rate compared to that of the latter.

**Definition 1** (Eigencapacity). *Let $p \in C[0,1]$ and $p(\mathbf{A})$ represent the filter kernel applied during the $n^{th}$ denoising step, as specified by Eqn. 13. The eigencapacity of this step, denoted by $\kappa_n(p)$, is defined by the integral*

$$\kappa_n(p) := \int_0^1 p^n(x)\,dx. \tag{15}$$

Note that $\kappa_n(p)$, which represents the area under the curve $p^n(x)$ over the interval $x \in [0,1]$, exhibits a strong correlation with (the sum of) the well-preserved magnitudes of the eigenvalues of $p^n(\mathbf{A})$ at iteration step $n$. This correlation arises because the integral of $p^n(x)$ over this range provides an effective approximation of this sum, particularly for matrices of considerable size since mean value theorem for definite integrals implies that

$$\frac{1}{N}\sum_{i=1}^{N} p^n(\lambda_i) \approx \int_0^1 p^n(x)\rho(x)dx = \rho(c)\kappa_n(p)$$

for some $c \in [0,1]$, where $\rho$ is a PDF of eigenvalue distribution. This observation underscores the integral's utility in approximating eigenvalue-related characteristics of the filter dyncamics represented by $p^n(\mathbf{A})$. In the following Proposition 1, we show that the eigencapacity of $2\mathbf{A} - 2\mathbf{A}^2$ decays at significantly slower rate than $\mathbf{A}$.

**Proposition 1** (Representational capacity decay rates). *Consider a denoising process employing the filter kernels $p(\mathbf{A}) = \mathbf{A}$ and $\hat{p}(\mathbf{A}) = 2\mathbf{A} - \mathbf{A}^2$. The eigencapacity $\kappa_n(\hat{p})$ decays at a rate of $\mathcal{O}(n^{-1/2})$, in contrast to $\mathcal{O}(n^{-1})$ for $\kappa_n(p)$. Specifically, the behavior of these eigencapacities as*

$n \to \infty$ *is given by:*

$$\kappa_n(p) \sim \frac{1}{n}, \tag{16}$$

$$\kappa_n(\hat{p}) \sim \frac{\sqrt{\pi}}{2\sqrt{n}}. \tag{17}$$

**Remark 1.** *Due to the equivalence that has been established between NLM smoothing and self-attention computation in Section 2.3, Proposition 1 demonstrates that if $2\mathbf{A} - \mathbf{A}^2$ was used as a similarity matrix in self-attention mechanism, the output would correspond to a nonlocal smoothing operation for which the convergence to a degenerate solution is significantly slower and, thus, capable of maintaining representational capacity for more iterations.*

We refer the reader to Appendix A.1 for the proof of Proposition 1.

**Relation to twicing kernels in nonparametric regression.** Equivalence between standard self-attention computation and Nadaraya-Watson estimator using isotropic Gaussian kernels in nonparametric regression has been established and used in numerous recent works (Nguyen et al., 2022c; Han et al., 2023; Nielsen et al., 2025). In particular, it has been shown that the output of a self-attention block is a discrete form of convolution of Gaussian kernel with bandwidth $\sqrt{D}$ and the value function (detailed in Appendix A.4). We reinterpret attention computation as $2\mathbf{A} - \mathbf{A}^2$ rather than $\mathbf{A}$ in the nonparametric regression setting. If multiplying by the attention matrix $\mathbf{A}$ is equivalent to using some kernel $K$ for NW estimation, then using $\mathbf{A}^2$ is equivalent to applying the convolved kernel $K * K$ instead of $K$ (see Appendix A.5).

Therefore, while standard self-attention computation implicitly performs Nadaraya-Watson estimator by employing the kernel $K$, attention computation with $2\mathbf{A} - \mathbf{A}^2$ is equivalent to employing the modified kernel $2K - K * K$, which is exactly the same as applying the *kernel twicing* procedure to the original regression kernel (Stuetzle & Mittal, 1979; Newey et al., 2004; Abdous, 1995). This constructs higher order kernels with small bias property (SBP) which refers to a kernel's ability to reduce the leading-order term in the bias of the estimator as demonstrated in Proposition 2 below.

**Proposition 2** (Twicing kernels reduce the estimator bias). *Let $K(u)$ be a symmetric kernel function used in the Nadaraya-Watson estimator with bandwidth $h$. Define a new kernel $\hat{k}(u)$ as*

$$\hat{K}(u) = 2K(u) - (K * K)(u),$$

*where $(K * K)(u)$ denotes the convolution of $K(u)$ with itself. Then, the kernel $\hat{K}(u)$ yields a Nadaraya-Watson estimator with a smaller bias than that using $K(u)$.*

**Remark 2.** *Due to the relation that has been established between $2\mathbf{A} - \mathbf{A}^2$ and $2K - K * K$, Proposition 2 implies that if $2\mathbf{A} - \mathbf{A}^2$ was used as a similarity matrix in self-attention mechanism, the output would correspond to a Nadaraya-Watson estimator with lower bias and arguably less sensitive to bandwidth selection (Newey et al., 2004). This reduced sensitivity mitigates the bias fluctuations often introduced by slight adjustments, making the attention mechanism inherently more resilient to adversarial perturbations and improve model's robustness (Chernozhukov et al., 2022).*

The proof of Proposition 2 is provided in Appendix A.2. For a comprehensive statistical discussion on the topic, we direct the reader to (Newey et al., 2004) and the references therein.

**Twicing kernels benefit from residuals.** Recall that we have established connection between self-attention matrices $\mathbf{A}$ and $2\mathbf{A} - \mathbf{A}^2$ to regression kernels $K$ and $2K - K * K$, respectively. Therefore, we use smoothing and computing the attention output interchangeably. Now we provide a core constructive difference between the two kernel computations. Given a kernel-type smoother $\mathbf{A}$ and observations $\mathbf{V}^\ell(x)$ at iteration $\ell$, twicing procedure takes the following three steps:

1. Smooth $\mathbf{V}^\ell(x)$ and obtain $\mathbf{A}\mathbf{V}^\ell(x)$.
2. Smooth the residual $\mathbf{V}^\ell(x) - \mathbf{A}\mathbf{V}^\ell(x)$ and obtain the correction $\mathbf{A}(\mathbf{V}^\ell(x) - \mathbf{A}\mathbf{V}^\ell(x)) = (\mathbf{A} - \mathbf{A}^2)\mathbf{V}^\ell(x)$.
3. Combine Step 1 and Step 2 and define $(2\mathbf{A} - \mathbf{A}^2)\mathbf{V}^\ell(x)$ as the new estimator.

Note that the final estimator actually consists of two terms: the first term corresponds to the denoised image via the filter $\mathbf{A}$, and the second term is the residual $\mathbf{V}^\ell - \mathbf{A}\mathbf{V}^\ell$, which is also smoothed with $\mathbf{A}$.

Table 1: Top-1 and Top-5 Test Accuracy on ImageNet corrupted by projected gradient descent (PGD), fast gradient sign method (FGSM), and simultaneous perturbation stochastic approximation (SPSA).

| Model | ImageNet | | PGD | | FGSM | | SPSA | |
|---|---|---|---|---|---|---|---|---|
| | Top 1 | Top 5 | Top 1 | Top 5 | Top 1 | Top 5 | Top 1 | Top 5 |
| *DeiT* (Touvron et al., 2021) | 72.00 | 91.14 | 8.16 | 22.37 | 29.88 | 63.26 | 66.41 | 90.29 |
| NeuTRENO (Nguyen et al., 2023b) | 72.44 | **91.39** | 8.85 | 23.83 | 31.43 | **65.96** | 66.98 | 90.48 |
| DeiT-Twicing [10-12] | 72.31 | 91.24 | 8.66 | 22.58 | 31.63 | 64.74 | 66.47 | 90.49 |
| DeiT-Twicing | **72.60** | 91.33 | **9.15** | **24.10** | **32.28** | 65.67 | **67.12** | **90.53** |
| *FAN* (Zhou et al., 2022) | 77.09 | 93.72 | 11.91 | 24.11 | 33.81 | 65.25 | 67.15 | 92.14 |
| FAN-Twicing | **77.18** | **94.02** | **12.80** | **28.86** | **35.52** | **67.23** | **68.89** | **93.75** |

Table 2: Evaluation of the performance of DeiT and DeiT-Twicing in ImageNet classification under the presence of different corruptions, using appropriate evaluation metrics for each.

| Dataset
Metric | ImageNet-R
Top 1 | ImageNet-A
Top 1 | ImageNet-C
mCE ($\downarrow$) | ImageNet-C (Extra)
mCE ($\downarrow$) |
|---|---|---|---|---|
| *DeiT* (Touvron et al., 2021) | 32.22 | 6.97 | 72.21 | 63.68 |
| DeiT-Twicing [10-12] | 32.31 | **8.14** | **70.25** | 62.63 |
| DeiT-Twicing | **32.74** | 7.66 | 70.33 | **62.46** |
| *FAN* (Zhou et al., 2022) | 42.24 | **12.33** | 60.71 | 52.70 |
| FAN-Twicing | **42.36** | 12.30 | **60.48** | **52.21** |

Therefore, denoising with kernel $\hat{p}(\mathbf{A})$ is equivalent to denoising with kernel $p(\mathbf{A})$ and subsequently feeding the smoothed method noise of this denoising step back into the output of the current iteration to effectively extracts salient information remaining in the residual and reincorporates it into the denoising output.

### 3.4 TWICING ATTENTION: FULL TECHNICAL FORMULATION

Stemming from the theoretical benefits discussed in the previous sections, we formulate Twicing Attention as follows:

**Definition 2** (Twicing Attention). *Given query* $\mathbf{Q}^\ell = [\boldsymbol{q}_1^\ell, \ldots, \boldsymbol{q}_N^\ell]^\top \in \mathbb{R}^{N \times D}$, *key* $\mathbf{K}^\ell = [\boldsymbol{k}_1^\ell, \ldots, \boldsymbol{k}_N^\ell]^\top \in \mathbb{R}^{N \times D}$, *and value* $\mathbf{V}^\ell = [\boldsymbol{v}_1^\ell, \ldots, \boldsymbol{v}_N^\ell]^\top \in \mathbb{R}^{N \times D}$ *matrices as in Section 2.1 at* $\ell^{th}$ *layer of transformer, the output of Twicing Attention mechanism is computed as:*

$$\mathbf{U}^\ell = (2\mathbf{A} - \mathbf{A}^2)\mathbf{V}^\ell, \tag{18}$$

*where* $\mathbf{A} \coloneqq \mathrm{softmax}\left(\mathbf{Q}^\ell \mathbf{K}^{\ell\top}/\sqrt{D}\right)$ *and the softmax function is applied row-wise.*

**Remark 3.** *Even though Definition 2 gives Twicing Attention computation as in Eqn. 18, we use the following equivalent, a twicing procedure-inspired form in practice:*

$$\mathbf{U}^\ell = \mathbf{A}\mathbf{V}^\ell + \mathbf{A}(\mathbf{V}^\ell - \mathbf{A}\mathbf{V}^\ell). \tag{19}$$

*In other words, instead of computing the square of attention matrix* $\mathbf{A}^2$, *we decompose Eqn. 18 into regular self-attention output and smoothed residual parts as* $\mathbf{A}\mathbf{V}^\ell + \mathbf{A}(\mathbf{V}^\ell - \mathbf{A}\mathbf{V}^\ell)$. *This allows us to compute* $\mathbf{A}\mathbf{V}^\ell$ *once and reuse it in the residual to replace attention squaring operation, which is* $\mathcal{O}(N^3)$, *with cheaper matrix multiplication of* $\mathcal{O}(N^2 D)$ *runtime complexity matching the standard self-attention computation.*

## 4 EXPERIMENTAL RESULTS

In this section, we empirically justify the advantage of Twicing Attention over baseline transformers with standard self-attention mechanism. Whenever we employ our Twicing Attention to replace the standard one in a given model, we append a *Twicing* suffix to imply this in the reports. Moreover, if Twicing Attention is inserted in specific transformer layers only, we specify the layer indices in square brackets ([10-12] for Twicing Attention in layers 10, 11 and 12, etc.). We evaluate our method

Table 3: Image segmentation on ADE20K.

| Model | Pix. Acc. | Mean Acc. | Mean IoU |
|---|---|---|---|
| *DeiT* | 77.25 | 44.48 | 34.73 |
| DeiT-Twicing | **77.51** | **45.53** | **35.12** |

Table 4: Test PPL on WikiText-103.

| Model | Test PPL | Attacked PPL |
|---|---|---|
| *Transformer* | 37.51 | 55.17 |
| Tr.-Twicing | **36.69** | **54.46** |

on Wikitext-103 modeling both under clean and Word Swap contamination (Merity et al., 2016), and ImageNet-1K classification under a wide range of attacks (Deng et al., 2009; Russakovsky et al., 2015) as described in detail in the following paragraphs.

## 4.1 IMAGE CLASSIFICATION AND SEGMENTATION

**Object classification on ImageNet-1K.** To demonstrate the advantage of our model, we compare it with the *DeiT* baseline (Touvron et al., 2021) and NeuTRENO (Nguyen et al., 2023b) on the Im-ageNet1K image classification task (Deng et al., 2009). Our model surpasses the DeiT baseline, as shown in Table 1 in the clean data setting as well as under adversarial attacks such as fast gradient sign method (FGSM) (Goodfellow et al., 2014), projected gradient descent (PGD) (Madry et al., 2017) with perturbation budget $4/255$ and provide a comparison of results for different values of perturbations in appendix, and simultaneous perturbation stochastic approximation (SPSA) (Uesato et al., 2018) with perturbation budget $1/255$. Furthermore, Table 2 shows DeiT-Twicing to be con-sistently more robust than the DeiT baseline across various testing conditions, including adversarial examples and out-of-distribution datasets. This includes its performance on the Imagenet-C dataset, which involves common data corruptions and perturbations such as noise addition and image blur-ring, as well as on Imagenet-A and Imagenet-R datasets, which assess adversarial example handling and out-of-distribution generalization, respectively (Hendrycks et al., 2021). ImageNet-C (Extra) contains four extra image corruption types: spatter, gaussian blur, saturate, and speckle noise.

When combining with a state-of-the-art robust transformer backbone, *Fully Attentional Network* (*FAN*) (Zhou et al., 2022), Twicing Attention is able to improve performance in terms of clean accu-racy as well as its robustness against adversarial attacks such as PGD and FGSM (with perturbation budget 4/255) as well as SPSA (with perturbation budget 1/255) substantially as included in Table 1. We also find better out-of-distribution generalization in FAN-Twicing over standard FAN except for ImageNet-A benchmark where FAN-Twicing is still highly competitive (see Table 2).

**Image segmentation on ADE20K.** On top of the classification task, we compare the performance of the Segmenter models using DeiT and DeiT-Twicing backbones on the ADE20K (Zhou et al., 2019) image segmentation task to further validate the advantages of our proposed method by adopting the experimental setup of (Strudel et al., 2021). In Table 3, we report the key metrics: pixel accuracy, mean accuracy, and mean intersection over union (IOU). We observe performance boost across all 3 metrics with DeiT-Twicing over the DeiT baseline (Touvron et al., 2021).

## 4.2 LANGUAGE MODELING ON WIKITEXT-103

In addition to computer vision tasks, we also evaluate the effectiveness of our model on a large-scale natural language processing application, specifically language modeling on WikiText-103 (Merity et al., 2016). Our language model demonstrates better performance in terms of both test perplexity (PPL) and valid perplexity when compared to the standard *transformer* language model (Vaswani et al., 2017) as shown in Table 4. We also show that test PPL on WikiText-103 contaminated by Word Swap Attack, where words are randomly swapped with a generic token 'AAA'. We follow the setup of (Han et al., 2023; Teo & Nguyen, 2025) and assess models by training them on clean data before attacking only the test data using an attack rate of 4%.

## 5 EMPIRICAL ANALYSIS

**Representation collapse analysis.** We empirically demonstrate in Figure 1 that Twicing Attention mechanism promotes better token diversity and, therefore, it is able to slow down representation collapse phenomenon in transformers. We observe that, in DeiT, the average cosine similarity score between tokens quickly exceeds three-quarters, whereas in our model, it remains consistently below this threshold. Additionally, we demonstrate in Figure 2 that our model is indeed capable of retaining better expressive power by being able to pay attention to notably more and important parts of objects in images while DeiT indicates collapsed behaviour. See Appendix D.2 for a dozen of additional attention heatmaps supporting the comparative argument.

Table 5: Efficiency comparison between DeiT and DeiT-Twicing models.

| Model | Avg. Compute Speed (ms/it) | GFLOPs / sample | Param. Count (M) |
|---|---|---|---|
| *DeiT* (Touvron et al., 2021) | 8.58 | 1.25 | 5.7 |
| DeiT-Twicing | 9.14 | 1.33 | 5.7 |
| DeiT-Twicing [10-12] | 8.72 | 1.27 | 5.7 |

**Efficiency analysis.** As stated in Remark 3, our Twicing Attention mechanism can be implemented with $\mathcal{O}(N^2 D)$ runtime complexity, which is on par with the standard self-attention mechanism. Table 5 compares the prediction average compute speed per iteration over 1000 runs as well as the floating-point operations (FLOPs) per sample, which is a measure of trade-off between efficiency (lower FLOPs) and accuracy (higher FLOPs) of models. We observe that employing Twicing Attention only in last 3 layers, the model can still enjoy performance gains over the baseline while seeing almost negligible increase in average compute speed and FLOPs per sample.

## 6 RELATED WORK

**Theoretical frameworks for attention.** Attention mechanisms have been studied from a range of perspectives. (Tsai et al., 2019) shows that attention can be derived from kernel similarity functions. (Nguyen et al., 2023d; Tarzanagh et al., 2023) relate attention to support vector regression/machine, while (Tao et al., 2023) explains attention through nonlinear singular value decomposition of asymmetric kernels. Furthermore, (Teo & Nguyen, 2025) establishes a connection between self-attention and kernel principal component analysis, demonstrating that self-attention maps query vectors onto the principal component axes of the key matrix in a feature space. Attention has also been explained through Gaussian mixture models, ordinary/partial differential equations, optimization algorithms, and graph-structured learning (Nguyen et al., 2022a;b; 2023c; Tang & Matteson, 2021; Gabbur et al., 2021; Lu et al., 2019; Sander et al., 2022; Nguyen et al., 2022d; Kreuzer et al., 2021; Zhang & Feng, 2021) or an energy functional minimization associated with a variational image denoising framework (Nguyen et al., 2023b). (Nguyen et al., 2022c; Han et al., 2023; Nielsen et al., 2025) show that self-attention performs Nadaraya-Watson regression with Gaussian isotropic kernels.

**Representation collapse in transformers.** Representation collapse or over-smoothing in transformers has been observed across various domains, including NLP (Shi et al., 2022) and computer vision (Wang et al., 2022). (Shi et al., 2022) analyzes this issue in BERT (Devlin et al., 2018) using a graph-based approach, employing hierarchical fusion techniques to retain self-attention outputs, though at a high memory cost. (Dong et al., 2021) is among the first to explore oversmoothing in transformers through rank collapse, while (Caron et al., 2021) examines self-supervised ViTs, revealing their embedded semantic segmentation information. Additionally, (Darcet et al., 2024) identifies high-norm token artifacts in these feature maps. Furthermore, (Nguyen et al., 2024a) explores the link between self-attention and the state-space model, showing that attention output collapses into a steady-state solution.

**Robust transformers.** For transformers, robust strategies include an ensemble defense against adversarial attacks (Mahmood et al., 2021), position-aware attention scaling with patch-wise augmentation (Mao et al., 2022), and a fully-attentional network for state-of-the-art performance on corrupted images (Zhou et al., 2022). Efficiency usually refers to optimal performance under ideal conditions, while robustness describes maintaining strong performance under less-than-ideal circumstances. A common trend among robust models, such as (Mao et al., 2022; Han et al., 2023; Zhou et al., 2022), is their reliance on additional computational overhead, often matching or even exceeding that of our proposed model.

## 7 CONCLUDING REMARKS

In this paper, we introduced the Twicing Attention mechanism, enhancing the transformer's representational capacity by utilizing residuals between self-attention inputs and outputs. This novel self-attention variant improves token diversity and mitigates representational collapse by leveraging useful residual information as a form of self-correction. We empirically demonstrated performance gains on ImageNet-1k, ADE20K, WikiText-103, and robustness benchmarks with minimal computational overhead by trying selective layer placement for Twicing Attention. However, limitations include its efficient application across transformer all layers with no or negligible additional computation. Ongoing work explores approximation techniques and sparsity to improve efficiency, while extending the theoretical framework to even more practical scenarios remains an open challenge.

ACKNOWLEDGMENTS

This research / project is supported by the National Research Foundation Singapore under the AI Singapore Programme (AISG Award No: AISG2-TC-2023-012-SGIL). This research / project is supported by the Ministry of Education, Singapore, under the Academic Research Fund Tier 1 (FY2023) (A-8002040-00-00, A-8002039-00-00). This research / project is also supported by the NUS Presidential Young Professorship Award (A-0009807-01-00) and the NUS Artificial Intelligence Institute–Seed Funding (A-8003062-00-00).

**Reproducibility Statement.** We have made efforts to ensure the reproducibility of our work through several measures. Source codes for our experiments are provided in the supplementary materials of the paper. The details of our experimental settings and computational infrastructure are given in Section 4 and the Appendix. All datasets that we used in the paper are published, and they are easy to find in the Internet. These resources and explanations should allow others to replicate our results with relative ease.

**Ethics Statement.** Given the nature of our work and contributions, we do not foresee any negative societal and ethical impacts of our work.

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

# Supplement to "Transformer Meets Twicing: Harnessing Unattended Residual Information"

**Table of Contents**

## A TECHNICAL PROOFS AND DERIVATIONS

### A.1 PROOF OF PROPOSITION 1

The equivalence in Eqn. 16 is straightforward to obtain since $\kappa_n(p)$ can be calculated as

$$\kappa_n(p) = \int_0^1 p^n(x)dx = \int_0^1 x^n dx = \frac{1}{n+1} \sim \frac{1}{n}.$$

To prove the equivalence given by Eqn. 17, we first observe that

$$\kappa_n(\hat{p}) = \int_0^1 \hat{p}^n(x)dx = \int_0^1 (2x - x^2)^n dx = \frac{1}{2}\int_0^2 (2x - x^2)^n dx,$$

where the last equality is due to the symmetry of $2x - x^2 = 1 - (1-x)^2$ along $x = 1$. Now, employing a variable change $x = 2y$ yields

$$\kappa_n(\hat{p}) = \frac{1}{2}\int_0^2 (2x - x^2)^n dx = \int_0^1 (4y - 4y^2)^n dy$$

$$= 4^n \int_0^1 y^n (1-y)^n dy = 4^n B(n+1, n+1) \tag{20}$$

$$= \frac{4^n \Gamma(n+1)^2}{\Gamma(2n+2)} = \frac{4^n (n!)^2}{(2n+1)!}, \tag{21}$$

where $B(x, y)$ and $\Gamma(x)$ denote the Euler Beta function and Gamma function, respectively, and we used the identity $B(x, y) = \Gamma(x)\Gamma(y)/\Gamma(x + y)$ to transform Eqn. 20 into Eqn.21. Now using

Stirling's approximation $n! \sim \sqrt{2\pi n}(n/e)^n$ as $n \to \infty$ for Eqn. 21, we obtain

$$
\begin{aligned}
\kappa_n(\hat{p}) &\sim \frac{4^n \cdot 2\pi n^{2n+1}/e^{2n}}{\sqrt{2\pi(2n+1)}(2n+1)^{2n+1}/e^{2n+1}} \\
&= e\sqrt{\frac{\pi}{2}} \frac{1}{\sqrt{2n+1}} \left(\frac{2n}{2n+1}\right)^{2n+1} \\
&= \frac{e\sqrt{\pi}}{2} \frac{1}{\sqrt{n+1/2}} \left(1 - \frac{1}{2n+1}\right)^{2n+1} \\
&\sim \frac{\sqrt{\pi}}{2\sqrt{n}},
\end{aligned}
\tag{22}
$$

where we used the fact that $e^{-1} = \lim_{n\to\infty} \left(1 - \frac{1}{2n+1}\right)^{2n+1}$ to derive Eqn. 22. $\qquad\square$

## A.2 PROOF OF PROPOSITION 2

*Proof of Proposition 2.* To compare the biases of the estimators using kernels $K(u)$ and $\hat{K}(u)$, we analyze the moments of these kernels, as they determine the bias in kernel estimators.

We begin with showing that $\hat{K}$ has valid kernel propoerties.

*Normalization.* Since $K(u)$ is a valid kernel, we have:

$$
\int_{\mathbb{R}} K(u)\, du = 1.
$$

The convolution of $K(u)$ with itself satisfies:

$$
\int_{\mathbb{R}} (K * K)(u)\, du = \left(\int_{\mathbb{R}} K(u)\, du\right)^2 = 1.
$$

Therefore,

$$
\int_{\mathbb{R}} \hat{K}(u)\, du = 2\int K(u)\, du - \int (K * K)(u)\, du = 1.
$$

Thus, $\hat{K}(u)$ is normalized.

*Symmetry.* If $K(u)$ is symmetric, i.e., $K(u) = K(-u)$, then $(K * K)(u)$ is also symmetric. Therefore,

$$
\hat{K}(-u) = 2K(-u) - (K * K)(-u) = 2K(u) - (K * K)(u) = \hat{K}(u).
$$

Thus, $\hat{K}(u)$ is symmetric.

*Zero First Moment.* The first moment of a kernel should be zero:

$$
\int_{\mathbb{R}} u\hat{K}(u)\, du = 2\int uK(u)\, du - \int u(K * K)(u)\, du.
$$

Since $K(u)$ is symmetric, $\int uK(u)\, du = 0$, and the convolution $(K * K)(u)$ is also symmetric, so $\int u(K * K)(u)\, du = 0$. Therefore,

$$
\int_{\mathbb{R}} u\hat{K}(u)\, du = 0.
$$

This confirms that $\hat{K}(u)$ has a zero first moment.

Next, note that the second moment of a kernel function, $\mu_2$, influences the leading term in the bias of the kernel estimator. For $\hat{K}(u)$, we have:

$$
\mu_2(\hat{K}) = \int_{\mathbb{R}} u^2 \hat{K}(u)\, du = 2\int u^2 K(u)\, du - \int u^2 (K * K)(u)\, du. \tag{23}
$$

We know that $\int u^2 K(u)\,du = \mu_2(K)$. The term $\int u^2(K * K)(u)\,du$ can be evaluated as follows:

$$
\begin{aligned}
\int u^2 (K * K)(u)\,du &= \int_{\mathbb{R}} u^2 \left( \int_{\mathbb{R}} K(v) K(u - v)\,dv \right) du \\
&= \int_{\mathbb{R}} K(v) \left( \int_{\mathbb{R}} u^2 K(u - v)\,du \right) dv \\
&= \int_{\mathbb{R}} K(v) \left( \int_{\mathbb{R}} (s + v)^2 K(s)\,ds \right) dv \\
&= \int_{\mathbb{R}} K(v) \left( \int s^2 K(s)\,ds + 2v \int s K(s)\,ds + v^2 \int K(s)\,ds \right) dv.
\end{aligned}
$$

Since $K(s)$ is symmetric:

$$
\int s K(s)\,ds = 0, \quad \int s^2 K(s)\,ds = \mu_2(K), \quad \int K(s)\,ds = 1.
$$

Thus, the expression simplifies to:

$$
\int_{\mathbb{R}} K(v)\left( \mu_2(K) + 0 + v^2 \cdot 1 \right) dv = \mu_2(K) \int K(v)\,dv + \int v^2 K(v)\,dv = 2\mu_2(K)
$$

Thus,

$$
\int u^2 (K * K)(u)\,du = \mu_2(K)(1) + \mu_2(K) = 2\mu_2(K).
$$

Finally, returning to $\mu_2(\hat{K})$ in Eqn. 23:

$$
\begin{aligned}
\mu_2(\hat{K}) &= 2 \int u^2 K(u)\,du - \int u^2 (K * K)(u)\,du \\
&= 2\mu_2(K) - 2\mu_2(K) \\
&= 0.
\end{aligned}
$$

*Implications for the bias.* A classical result in statistics imply that the leading bias term of the Nadaraya-Watson estimator using kernel $K(u)$ is proportional to $\mu_2(K)h^2$:

$$
\text{Bias}[\hat{m}_K(x)] \approx \frac{h^2}{2} \mu_2(K) m''(x),
$$

where $m''(x)$ is the second derivative of the true regression function at point $x$ (see, for example, (Wand & Jones, 1995)).

For the estimator using $\hat{K}(u)$, since $\mu_2(\hat{K}) = 0$, the leading bias term of order $h^2$ disappears. The next non-zero term in the bias expansion involves the fourth moment $\mu_4(\hat{K})$, resulting in a bias of order $h^4$:

$$
\text{Bias}[\hat{m}_{\hat{K}}(x)] \approx \frac{h^4}{24} \mu_4(\hat{K}) m^{(4)}(x).
$$

This demonstrates that the estimator using $\hat{K}(u)$ reduces leading order bias terms that appear when $K(u)$ is used. $\qquad\square$

### A.3 DERIVATION OF GRADIENT OF $J_\omega$

Expand the functional $J_\omega(\boldsymbol{u})$ as follows:

$$
J_\omega(\boldsymbol{u}) = \frac{1}{2} \int_{\Omega \times \Omega} \sum_{j=1}^{D} \left( u_j(x) - u_j(y) \right)^2 w(x, y)\,dx\,dy \tag{24}
$$

The gradient of $J_\omega$ with respect to $\boldsymbol{u}$ is then given by:

$$
\nabla_u J_\omega(u) = \left[ \frac{\partial J_\omega}{\partial u_1}, \frac{\partial J_\omega}{\partial u_2}, \ldots, \frac{\partial J_\omega}{\partial u_D} \right]^\top \tag{25}
$$

The partial derivative $\partial J_\omega / \partial u_j$, for $j = 1, 2, \ldots, D$, is defined through its dot product with an arbitrary function $h_j \in L^2(\Omega)$ as follows:

$$
\begin{aligned}
\frac{\partial J_\omega}{\partial u_j} \cdot h_j(x) &= \frac{d}{d\tau} J_\omega(u_j + \tau h_j)\bigg|_{\tau=0} \\
&= \frac{1}{2}\left(\frac{d}{d\tau} \int_{\Omega \times \Omega} (u_j(x) - u_j(y) + \tau h_j(x) - \tau h_j(y))^2 \, w(x,y) dx dy\right)\bigg|_{\tau=0} \\
&= \frac{1}{2}\left(\frac{d}{d\tau} \int_{\Omega \times \Omega} (u_j(x) - u_j(y) + \tau h_j(x) - \tau h_j(y))^2 \, w(x,y) dx dy\right)\bigg|_{\tau=0} \\
&= \int_{\Omega \times \Omega} (u_j(x) - u_j(y))(h_j(x) - h_j(y)) \, w(x,y) dx dy
\end{aligned}
$$

Applying a change of variables $(x, y) \to (y, x)$ to the second term of the above integral, we have:

$$
\frac{\partial J_\omega}{\partial u_j} \cdot h_j(x) = \int_\Omega (u_j(x) - u_j(y)) h_j(x) (w(x,y) + w(y,x)) \, dy
$$

Thus, the Frechet derivative of $J_\omega$ with respect to $u_j$ is given by:

$$
\frac{\partial J_\omega}{\partial u_j} = \int_\Omega (u_j(x) - u_j(y))(k(x,y) + k(y,x)) \, dy, \tag{26}
$$

which then gives the desired gradient with $w(x,y) \leftarrow w(x,y) + w(y,x)$ (Nguyen et al., 2023b).

## A.4 Equivalence of Self-attention and Nadaraya-Watson Estimator

We establish the relationship between self-attention, as defined in Eqn. 1, and non-parametric regression following the approaches of (Nguyen et al., 2022c; Han et al., 2023; Nielsen et al., 2025). To begin, let us assume that the key and value vectors $\{\boldsymbol{k}_j, \boldsymbol{v}_j\}_{j \in [N]}$ are generated by the following data process:

$$
\boldsymbol{v} = f(\boldsymbol{k}) + \boldsymbol{\epsilon}, \tag{27}
$$

where $\boldsymbol{\epsilon}$ represents random noise with zero mean, i.e., $\mathbb{E}[\epsilon] = 0$, and $f$ is the unknown function we aim to estimate. In this setup, the keys $\{\boldsymbol{k}_j\}_{j \in [N]}$ are independent and identically distributed (i.i.d.) samples drawn from the marginal distribution $p(\boldsymbol{k})$, characterizing the random design setting. We use $p(\boldsymbol{v}, \boldsymbol{k})$ to denote the joint distribution of the pairs $(\boldsymbol{v}, \boldsymbol{k})$ generated by the process described in Eqn. 27. For a new query $\boldsymbol{q}$, our goal is to estimate the function $f(\boldsymbol{q})$.

Recall NW estimator is a non-parametric estimator of the unknown $f$ at any given query $\boldsymbol{q}$ described by

$$
f(\boldsymbol{k}) = \mathbb{E}[\boldsymbol{v} \mid \boldsymbol{k}] = \int_{\mathbb{R}^D} \boldsymbol{v} \cdot p(\boldsymbol{v} \mid \boldsymbol{k}) d\boldsymbol{v} = \int_{\mathbb{R}^D} \frac{\boldsymbol{v} \cdot p(\boldsymbol{v}, \boldsymbol{k})}{p(\boldsymbol{k})} d\boldsymbol{v},
$$

where the first equality comes from the noise being zero mean, the second equality comes from the definition of conditional expectation and the final equality comes from the definition of conditional density. Eqn. 27 implies that if we can just obtain good estimates of the joint density $p(\boldsymbol{v}, \boldsymbol{k})$ and marginal density $p(\boldsymbol{k})$ then we can estimate the required $f(\boldsymbol{q})$. The Gaussian isotropic kernels with bandwidth $\sigma$ are given by

$$
\hat{p}_\sigma(\boldsymbol{v}, \boldsymbol{k}) = \frac{1}{N} \sum_{j \in [N]} \varphi_\sigma(\boldsymbol{v} - \boldsymbol{v}_j) \varphi_\sigma(\boldsymbol{k} - \boldsymbol{k}_j), \quad \hat{p}_\sigma(\boldsymbol{k}) = \frac{1}{N} \sum_{j \in [N]} \varphi_\sigma(\boldsymbol{k} - \boldsymbol{k}_j), \tag{28}
$$

where $\varphi_\sigma$ is the multivariate Gaussian density function with diagonal covariance matrix $\sigma^2 \boldsymbol{I}_D$. Given the kernel density estimators in Eqn. 28, the unknown function can be estimated as

$$
\begin{aligned}
\hat{f}_\sigma(\boldsymbol{k}) &= \int_{\mathbb{R}^D} \frac{\boldsymbol{v} \cdot \hat{p}_\sigma(\boldsymbol{v}, \boldsymbol{k})}{\hat{p}_\sigma(\boldsymbol{k})} \, d\boldsymbol{v} = \int_{\mathbb{R}^D} \frac{\boldsymbol{v} \cdot \sum_{j \in [N]} \varphi_\sigma(\boldsymbol{v} - \boldsymbol{v}_j) \varphi_\sigma(\boldsymbol{k} - \boldsymbol{k}_j)}{\sum_{j \in [N]} \varphi_\sigma(\boldsymbol{k} - \boldsymbol{k}_j)} \, d\boldsymbol{v} \\
&= \frac{\sum_{j \in [N]} \varphi_\sigma(\boldsymbol{k} - \boldsymbol{k}_j) \int \boldsymbol{v} \cdot \varphi_\sigma(\boldsymbol{v} - \boldsymbol{v}_j) d\boldsymbol{v}}{\sum_{j \in [N]} \varphi_\sigma(\boldsymbol{k} - \boldsymbol{k}_j)} = \frac{\sum_{j \in [N]} \boldsymbol{v}_j \varphi_\sigma(\boldsymbol{k} - \boldsymbol{k}_j)}{\sum_{j \in [N]} \varphi_\sigma(\boldsymbol{k} - \boldsymbol{k}_j)}.
\end{aligned}
$$

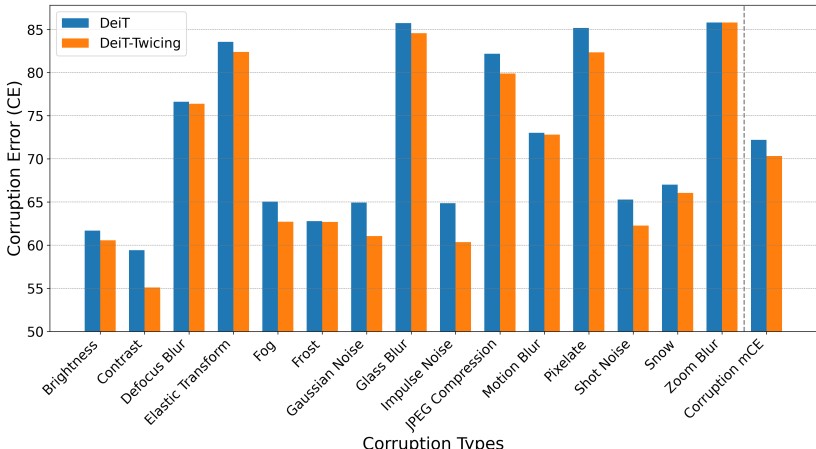

Figure 4: ImageNet-C corruption error (CE) (↓) and mean CE (mCE) (↓) comparison of our model and DeiT across all corruption types. Our model consistently outperforms DeiT.

Then, using the definition of the Gaussian isotropic kernel and evaluating the estimated function at $\boldsymbol{q}_i$ we have

$$
\begin{aligned}
\hat{f}(\boldsymbol{q}_i) &= \frac{\sum_j^N \boldsymbol{v_j} \exp\left(-\|\boldsymbol{q}_i - \boldsymbol{k}_j\|^2/2\sigma^2\right)}{\sum_j^N \exp\left(-\|\boldsymbol{q}_i - \boldsymbol{k}_j\|^2/2\sigma^2\right)} \\
&= \frac{\sum_j^N \boldsymbol{v_j} \exp\left[-(\|\boldsymbol{q}_i\|^2 + \|\boldsymbol{k}_j\|^2)/2\sigma^2\right]\exp(\boldsymbol{q}_i^\top \boldsymbol{k}_j/\sigma^2)}{\sum_j^N \exp\left[-(\|\boldsymbol{q}_i\|^2 + \|\boldsymbol{k}_j\|^2)/2\sigma^2\right]\exp(\boldsymbol{q}_i^\top \boldsymbol{k}_j/\sigma^2)} \\
&= \frac{\sum_j^N \boldsymbol{v_j} \exp(\boldsymbol{q}_i^\top \boldsymbol{k}_j/\sigma^2)}{\sum_j^N \exp(\boldsymbol{q}_i^\top \boldsymbol{k}_j/\sigma^2)} = \sum_{j=1}^N \mathrm{softmax}(\boldsymbol{q}_i^\top \boldsymbol{k}_j/\sigma^2)\boldsymbol{v}_j.
\end{aligned}
$$

as desired.

### A.5 EQUIVALENCE BETWEEN SELF-CONVOLUTION AND SQUARE OF ATTENTION MATRIX

Let $K$ denote the isotropic Gaussian kernel with bandwidth $h$. Then,

$$
\begin{aligned}
(K * K * \boldsymbol{v})(x) &= \int_\Omega K(x-t)(K * \boldsymbol{v})(t)\,dt = \int_\Omega K(x-t)\int_\Omega K(t-y)\boldsymbol{v}(y)\,dy\,dt \\
&= \int_\Omega \int_\Omega K(x-t)K(t-y)\,dt\,\boldsymbol{v}(y)\,dy \approx \int_\Omega \sum_{l=1}^N K(x-l)K(l-y)\boldsymbol{v}(y)\,dy \\
&\approx \sum_{j=1}^N \sum_{l=1}^N K(x-l)K(l-j)\boldsymbol{v}(j).
\end{aligned}
\tag{30}
$$

Taking $\mathbf{A}$ to be the NLM matrix whose entries are given by $\mathbf{A}_{ij} = w(i,j) = K(i-j)$, it becomes evident that Eqn. 30 can be represented as

$$
(K * K * \boldsymbol{v})(i) \approx \sum_{j=1}^N (\mathbf{A}^2)_{ij}\boldsymbol{v}(j).
\tag{31}
$$

## B  EXPERIMENTAL DETAILS AND ADDITIONAL RESULTS

### B.1  WIKITEXT-103 LANGUAGE MODELLING

**Dataset.** The WikiText-103[1] (WT-103) dataset contains around 268K words and its training set consists of about 28K articles with 103M tokens. This corresponds to text blocks of about 3600 words. The validation set and test sets consist of 60 articles with 218K and 246K tokens respectively.

[1] www.salesforce.com/products/einstein/ai-research/the-wikitext-dependency-language-modeling-dataset/

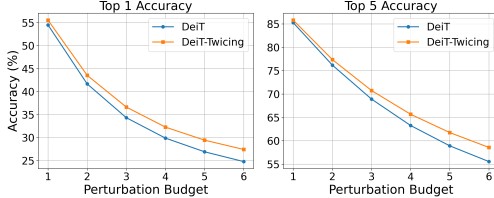 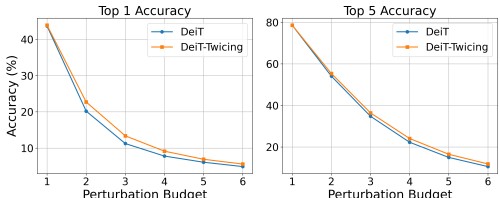

Figure 5: Top-1 and Top-5 accuracies on FGSM attack with 6 increasingly different perturbation budgets ($\times 255$).

Figure 6: Top-1 and Top-5 accuracies on PGD attack with 6 increasingly different perturbation budgets ($\times 255$).

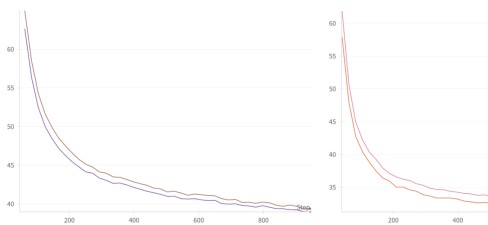

Figure 7: Validation PPL ($\downarrow$) training curves for baseline Transformer (higher) and Transformer-Twicing (lower). **Left:** small models (9.4M); **Right:** medium models (21M). We observe relatively faster convergence for Twicing Attention compared to standard self-attention.

**Corruption.** Word Swap Text Attack[2] corrupts the data by substituting random words with a generic token "AAA". We follow the setup of (Han et al., 2023) and assess models by training them on clean data before attacing only the evaluation set using a substitution rate of 4%.

**Model, Optimizer & Train Specification.** We adopt the training regime of (Nguyen et al., 2022c). To this end, the small backbone uses 16 layers, 8 heads of dimension 16, a feedforward layer of size 2048 and an embedding dimension of 128. We use a dropout rate of 0.1. We trained with Adam using a starting learning rate of 0.00025 and cosine scheduling under default PyTorch settings. We used a batch size of 96 and trained for 120 epochs and 2000 warmup steps. The train and evaluation target lengths were set to 256.

**Larger Language Modeling.** To verify if Twicing Attention scales, we conduct extra language modeling on Wikitext-103 with medium sized models (21M parameters) on top of the small models (9.4M parameters) reported in the main text. The results in Figure 7 and Table 6 imply a positive answer to this matter.

**Fine-tuning a pre-trained language model.** To show how Twicing Attention can offer improvements to the pre-trained models, we pre-train a medium sized (33M parameters) Switch Transformer (Fedus et al., 2022), a Mixture of Experts architecture, with the standard self-attention on WikiText-103. Then we finetune this pretrained language model on Stanford Sentiment Treebank 2 (SST-2) dataset using standard self-attention (baseline) as well as Twicing Attention (ours) for 8 epochs. Table 7 compares Top 1 fine-tune test accuracies for both cases and we find that fine-tuning with Twicing Attention achieves higher accuracy, provided that the fine-tuning is long enough for the model to adapt to the new attention mechanism.

Table 6: Valid/Test PPL on WT-103.

| Model | Valid PPL | Test PPL |
|---|---|---|
| *Transformer* (small) | 38.11 | 37.51 |
| Tr.-Twicing (small) | **37.12** | **36.69** |
| *Transformer* (med) | 31.98 | 26.17 |
| Tr.-Twicing (med) | **30.90** | **25.65** |
| *Linear Trans.* | 40.00 | 41.26 |
| Linear-Twicing | **39.45** | **40.61** |

Table 7: Fine-tuning Switch Transformer on SST-2.

| Attention Type | Test Acc. | #Params |
|---|---|---|
| *Self-attention* | 77.78 | 33M |
| Twicing Attention | **78.34** | 33M |

Table 8: Attacked Test PPL on WT-103.

| Model | Clean PPL | Attacked PPL |
|---|---|---|
| *Linear Trans.* | 41.26 | 72.13 |
| Linear-Twicing | **40.61** | **70.40** |

---

[2]Implementation available at github.com/QData/TextAttack

Table 9: Top-1 and Top-5 Test Accuracy on ImageNet corrupted by projected gradient descent (PGD), fast gradient sign method (FGSM), and simultaneous perturbation stochastic approximation (SPSA).

| Model | ImageNet | | PGD | | FGSM | | SPSA | |
|---|---|---|---|---|---|---|---|---|
| | Top 1 | Top 5 | Top 1 | Top 5 | Top 1 | Top 5 | Top 1 | Top 5 |
| *DeiT* (Touvron et al., 2021) | 72.00 | 91.14 | 8.16 | 22.37 | 29.88 | 63.26 | 66.41 | 90.29 |
| DeiT-Twicing [1-12] | **72.60** | 91.33 | **9.15** | **24.10** | **32.28** | **65.67** | **67.12** | **90.53** |
| DeiT-Twicing [7-12] | 72.45 | **91.35** | 8.67 | 22.90 | 31.60 | 64.79 | 66.48 | 90.52 |
| DeiT-Twicing [10-12] | 72.31 | 91.24 | 8.66 | 22.58 | 31.63 | 64.74 | 66.47 | 90.49 |

Table 10: Evaluation of the performance of DeiT and DeiT-Twicing in ImageNet classification under the presence of different corruptions, using appropriate evaluation metrics for each.

| Dataset | ImageNet-R | ImageNet-A | ImageNet-C | ImageNet-C (Extra) |
|---|---|---|---|---|
| Metric | Top 1 | Top 1 | mCE ($\downarrow$) | mCE ($\downarrow$) |
| *DeiT* (Touvron et al., 2021) | 32.22 | 6.97 | 72.21 | 63.68 |
| NeuTRENO (Nguyen et al., 2023b) | 31.65 | **8.36** | 70.51 | 63.56 |
| DeiT-Twicing | **32.74** | 7.66 | 70.33 | 62.46 |
| DeiT-Twicing [7-12] | 32.68 | 8.10 | **69.98** | **62.35** |
| DeiT-Twicing [10-12] | 32.31 | 8.14 | 70.25 | 62.63 |

**Non-conventional attention mechanisms.** To further validate the broad applicability of Twicing Attention (or twicing procedure in general), we conduct additional experiments following the theoretical setup of Linear Transformers (Katharopoulos et al., 2020). To be more precise, Table 6 (last 2 rows) compares the perplexities recorded for Linear Transformers with feature map $\phi(x) = \text{elu}(x) + 1$ matching their original choice, and Linear-Twicing Transformers for which we apply the twicing transformation $2\mathbf{A} - \mathbf{A}^2$ where $\mathbf{A} = \text{normalize}(\phi(\mathbf{Q})\phi(\mathbf{K})^\top)$ trained for 75 epochs, while Table 8 shows that Linear-Twicing model has superior robustness against the Word Swap Attack. Note that we explicitly construct the similarity matrix $A$ for both of the models for our framework to work. The positive results indicate that the applicability of Twicing Attention is not limited to standard softmax self-attention, but any reasonable similarity matrix can be covered.

## B.2 IMAGENET IMAGE CLASSIFICATION AND ADVERSARIAL ATTACK

**Dataset.** We use the full ImageNet dataset that contains 1.28M training images and 50K validation images. The model learns to predict the class of the input image among 1000 categories. We report the top-1 and top-5 accuracy on all experiments.

**Corruption.** We use attacks FGSM (Goodfellow et al., 2014) and PGD (Madry et al., 2017) with perturbation budget 4/255 while SPSA (Uesato et al., 2018) uses a perturbation budget 1/255. All attacks perturb under $l_\infty$ norm. PGD attack uses 20 steps with step size of 0.15.

**Model, Optimizer & Train Specification.** The configuration follows the default DeiT tiny configuration (Touvron et al., 2021). In particular, we follow the experimental setup of (Han et al., 2023; Nguyen et al., 2022c). To this end, the DeiT backbone uses 12 layers, 3 heads of dimension 64, patch size 16, feedforward layer of size 768 and embedding dimension of 192. We train using Adam with a starting learning rate of 0.0005 using cosine scheduling under default PyTorch settings, momentum of 0.9, batch size of 256, 5 warmup epochs starting from 0.000001 and 10 cooldown epochs, for an overall train run of 300 epochs. The input size is 224 and we follow the default AutoAugment policy and color jitter 0.4.

**Extra results.** In Figure 5 and Figure 6, we report that our model consistently outperforms the baseline across increasing six levels of severity under FGSM and PGD attacks. Additionally, Table 11 compares our model versus FeatScale (Wang et al., 2022), a vision transformer addressing oversmoothing. Our model outperforms in both metrics on clean ImageNet classification.

Table 11: ImageNet Classification.

| Model | Top 1 | Top 5 |
|---|---|---|
| *DeiT* | 72.00 | 91.14 |
| DeiT-FeatScale | 72.35 | 91.23 |
| DeiT-Twicing | **72.60** | **91.33** |

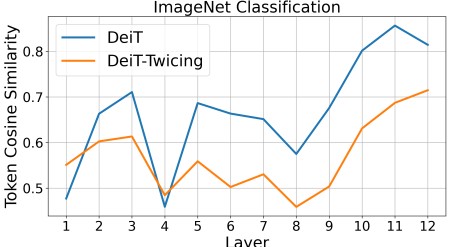
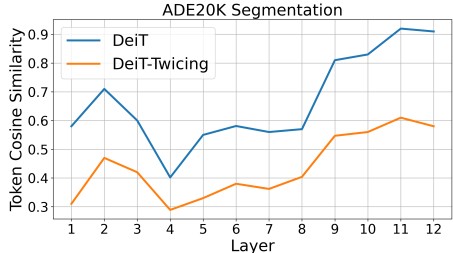

Figure 8: Comparison of average token cosine similarities across layers for DeiT and DeiT-Twicing models. Subfigure (a) uses ImageNet, while subfigure (b) evaluates ADE20K segmentation.

### B.3 Out-of-Distribution Robustness and Data Corruption on ImageNet-A,R,C

ImageNet-A,R,C are benchmarks capturing a range of out-of-distribution and corrupted samples (Hendrycks et al., 2021). ImageNet-A contains real world adversarially filtered images that fool current ImageNet classifiers. ImageNet-R contains various artistic renditions of object classes from the original ImageNet. ImageNet-C consists of 15 types of algorithmically generated corruptions with 5 levels of severity. (e.g blurring, pixelation, speckle noise etc). Figure 4 shows that DeiT-Twicing (our) model outperforms DieT baseline in all 15 main types of ImageNet-C corruptions.

### B.4 ADE20K Image Segmentation

**Experimental setup.** We adopt the setup in Strudel et al. (2021). The encoder is pretrained on ImageNet-1K following the same specification described in Appendix B.2. In particular, the encoder is a DeiT-tiny backbone of 5.7M parameters pretrained for 300 epochs. After pretraining, we then attach a decoder that contains 2-layer masked transformer and finetune the full encoder-decoder model for 64 epochs on the ADE20K Zhou et al. (2019) image segmentation dataset.

## C Compute Resources

**Training.** All models are trained using four NVIDIA A100 SXM4 40GB GPUs including both language and vision models.

**Evaluation.** Imagenet Classification under adversarial attacks are evaluated using two NVIDIA A100 SXM4 40GB GPUs while only one of such GPUs was used to evaluate against ImageNet-A,R,C and Word Swap Attack for language modelling.

## D Additional Empirical Analysis

### D.1 Extra Over-smoothing Analysis

We show the layer-wise over-smoothing analysis on both ImageNet classification and ADE20K image segmentation in Figure 8. We observe that in both cases, average token cosine similarities with Twicing Attention grow slower than those with standard self-attention, once again validating our theoretical findings.

### D.2 Extra Attention Heatmap Analysis

Extending the visualizations in the main text, Figure 9 illustrates how attention heatmaps evolve from early to late layers for DeiT and DeiT-Twicing models given the input images. In Figure 10, we provide 12 more examples to show that when employed Twicing Attention, DeiT-Twicing is usually more accurate to recognize objects in images and shows substantially better expressive power by capturing more meaningful parts of objects without missing target.

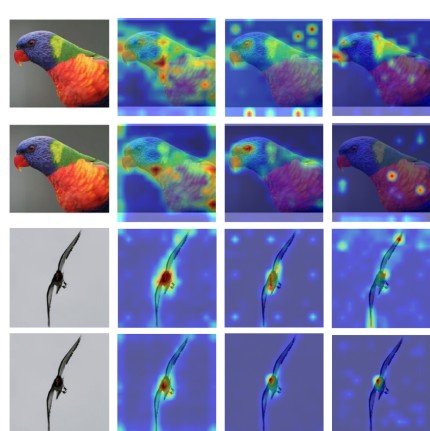

Figure 9: Evolution of attention heatmaps from early to late layers. **Odd rows:** DeiT-Twicing; **Even rows:** DeiT.

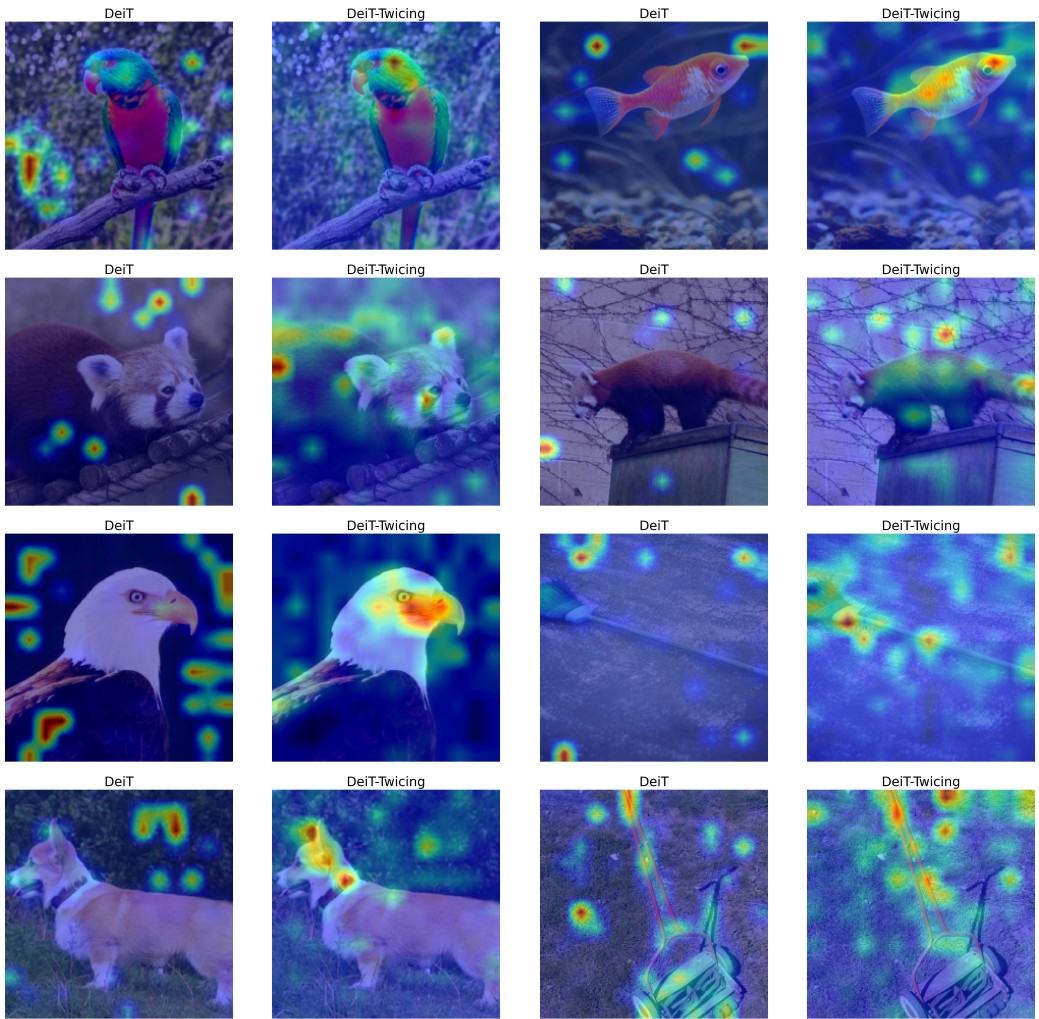

Figure 10: More examples showing how Twicing Attention is capable of retaining model's expressive power. The DeiT baseline model frequently collapses the entire image into the background, particularly when the background occupies a significant portion of the image, making it challenging to distinguish object details. Only in few cases, such as the example in bottom right, trying to capture more information was not as successful while still being highly competitive.

# E  ABLATION STUDIES

Since our proposed method does not require any additional parameters nor learnable, neither tunable, we only study the layer placement for Twicing Attention. Table 9 demonstrates 3 different layer placements of Twicing Attention - 1 to 12 (full), 7 to 12 (last six), and 10 to 12 (last three). We find that as long as Twicing Attention is placed starting from later layer till the end, the performance improvements are almost always proportional to the number of layers employing Twicing Attention. In Table 10, however, we observe that this proportionality does not happen in general since all three types of layer placements lead to good results in different categories, but all beating the baseline by notable margins. We also find that putting Twicing Attention only in few inital layers may not offer significant overall improvements.

