# OpenReview forum: "Transformer Meets Twicing: Harnessing Unattended Residual Information"
_ICLR.cc/2025/Conference — ICLR 2025 Poster_

### Official Review · Reviewer_7QSP · 2024-11-01

**Soundness:** 3
**Presentation:** 3
**Contribution:** 2
**Rating:** 5
**Confidence:** 3

**Summary:**

This paper introduces the Twicing Attention mechanism, drawing inspiration from established connections between self-attention and low-pass non-local means (NLM) smoothing filters. The authors demonstrate two key advantages of their proposed method: 1) a theoretically proven slower decay of representational capacity across transformer layers, and 2) improved performance on both vision and language tasks across multiple datasets. The paper's primary contribution lies in its theoretical framework. It first establishes that representation collapse in transformers stems from the inherent low-pass characteristics of NLM filters. The authors then provide proof showing that the twicing formulation ($2A^2-A$) offers superior theoretical properties compared to standard attention ($A$), particularly in preserving token diversity and meaningful feature representations.

**Strengths:**

- Theoretical foundation: The paper provides analysis connecting self-attention to NLM filters.
- Fluent presentation flow: Messages of this paper are presented well, with well-demonstrated background knowledge and motivation.
- Empirical validation: The authors provide visualizations of attention heatmaps, which validates their claim that their method preserve the diversity of token representations

**Weaknesses:**

- **Narrow Problem Framing**: The paper's central premise regarding "representation collapse" in transformers warrants closer scrutiny. Recent research has demonstrated that this phenomenon is not inherent to transformer architectures. For instance, DINO(Caron et al., 2021) demonstrates that self-supervised training can produce well-structured, diverse token representations in Vision Transformers. Furthermore, Darcet et al. (2024) provide evidence that apparent "collapse" may actually reflect a more nuanced information distribution pattern, where artifacts in attention heatmaps encode global information while non-artifact tokens maintain distinct representations, albeit with lower similarity to the CLS token.
- **Additional computational cost and marginal empirical improvements**: Performance increase in Table 4 is in trade of computational cost. Hardly can engineers be convinced to substitute the original attention with the proposed one.
- **Limited Evaluation Scope**: The authors report the empirical performance on classification tasks for vision models. Yet dense tasks such as segmentation are more direct and effective in evaluating the structure of patch representations produced by the method.

**Questions:**

- Visualizations on earlier layers and more heads of the transformers would help to strengthen your claim.
- Please refer to the weakness.
- I am open to increase my score if you alleviate my concerns.

---

> ### Author Response · Authors · 2024-11-20
> **Official Comment by Authors (1/2)**
>
> Thank you for your thoughtful review and valuable feedback. Below we address your concerns.
>
> -----
>
>
> **Q1. Narrow Problem Framing: The paper's central premise regarding "representation collapse" in transformers warrants closer scrutiny. Recent research has demonstrated that this phenomenon is not inherent to transformer architectures. For instance, DINO(Caron et al., 2021) demonstrates that self-supervised training can produce well-structured, diverse token representations in Vision Transformers. Furthermore, Darcet et al. (2024) provide evidence that apparent "collapse" may actually reflect a more nuanced information distribution pattern, where artifacts in attention heatmaps encode global information while non-artifact tokens maintain distinct representations, albeit with lower similarity to the CLS token.**
>
> **Answer:** We find the two papers that the reviewer brings to our attention interesting in terms of characterizing and understanding the emergent artifacts in the feature maps. Intriguingly, interpreting artifacts as locations where the model stores global input information—elegantly handled with extra register tokens in Darcet et al. (2024)—aligns (in a weak sense) with an image denoising perspective as well. When weighted averaging (blurring) is repeatedly applied, sharp edges are smoothed out, letting global information coming from large background sections dominate the output image. We note that the twicing procedure [3, 4] is tailored to guide the model to benefit from those local information and details before proceeding with another blurring iteration to accomodate both local and global information flow.
>
> On the other hand, there are at least a few fundamental scope differences between the  cited papers and ours, and our subject of study is not limited to representation collapse: (i) we mainly focus on the interpretation of our method through the perspective of twicing procedure and its analytical and statistical properties; (ii) while slowing down the decay of representational capacity is one of our contribution, it is not the only one. We believe the theoretical relation to twicing kernels with small bias property and its implications on learning more stable and robust representations is equally important matter of our paper; (iii) Unlike some prior works trying to mitigate over-smoothing completely by constantly fusing with initial layer tokens, we merely aim to slow it down to balance the mitigation of this phenomenon and largely deviating from the native behaviour of transformers to benefit from both worlds. All that being said, it is interesting to note how Twicing Attention heatmaps are usually concentrated over the body of the object while reducing the abovementioned artifacts as shown in a dozen of more sample images in Figure 11 in the appendix. Lastly, attention heatmaps are not the only way we illustrate "collapse", but we observe that, with Twicing Attention, average token similarities across layers indeed increase slower than the baseline as shown in Figure 2, which complements the other visualizations to validate slower collapse. Also, please see newly added Figure 8 in the appendix for both ImageNet and ADE20K oversmoothing analysis as a validation of our theoretical results on slower collapse.
>
> **References**
>
> [3]: Tukey, J.W. (1977). "Exploratory Data Analysis". Reading, MA: Addison-Wesley.
>
> [4]: Stuetzle, W., and Y. Mittal (1979): "Some Comments on the Asymptotic Behavior of Robust Smoothers", in Smoothing Techniques for Curve Estimation, Lecture Notes, 757. New York: Springer-Verlag, 191–195.

---

> > ### Author Response · Authors · 2024-11-20
> > **Official Comment by Authors (2/2)**
> >
> > **Q2. Additional computational cost and marginal empirical improvements: Performance increase in Table 4 is in trade of computational cost. Hardly can engineers be convinced to substitute the original attention with the proposed one.**
> >
> > **Answer:** Even though Twicing Attention offers relatively modest accuracy improvements in clean data settings, we believe that the complimentary robustness comparisons make the Twicing models stand out as substantially better models overall. In particular, Twicing models show capability to offer up to ~19\% improvement (FAN, PGD) with average of about ~8\% performance gains across all attacks. Besides, Figure 4 in the appendix shows that Twicing Attention can notably outperform the baseline across 15 types of natural corruption types consistently (about ~10% improvement on "contrast", "gaussian noise", and "impulse noise" to name but a few). It is worth noting that even many tailored robust models available with similar performances also impose similar additional computational cost. Additionally, this empirical observation is also theoretically consistent and interesting in the following sense: it suggests that Twicing models are inherently more stable across both clean and corrupted data settings by prioritizing stable representations over being specialized too much on clean data accuracy—an aspect that can make models more susceptible to small perturbations, such as common adversarial attacks. Additionally, drawing on the small-bias property of Twicing kernels in nonparametric regression, one can argue that the resulting estimator is relatively less sensitive to bandwidth selection. This reduced sensitivity mitigates the bias fluctuations often introduced by slight perturbations, making the estimator inherently more resilient to minor perturbations and improve its reliability. Our experimental results under 3 widely adopted adversarial attacks validate that Twicing Attention is indeed significantly more robust compared to the baseline self-attention.
> >
> > **Q3. Limited Evaluation Scope: The authors report the empirical performance on classification tasks for vision models. Yet dense tasks such as segmentation are more direct and effective in evaluating the structure of patch representations produced by the method.**
> >
> > **Answer:** Thank you for your feedback emphasizing the importance of evaluating our method on dense tasks like image segmentation to better assess patch representations. In response to your suggestion, we have conducted additional experiments on image segmentation and report the results in the table below and in Table 3 of the paper:
> >
> >  **Table G:** Image Segmentation on ADE20K.
> > | Model | Pixel Acc. | Mean Acc. | Mean IoU |
> > |-------|------------|-----------|----------|
> > | DeiT  | 77.25      | 44.48     | 34.73    |
> > | DeiT-Twicing | **77.51** | **45.53** | **35.12**
> >
> > These results indicate that our proposed DeiT-Twicing method offers improvements across key segmentation metrics, including Pixel Accuracy, Mean Accuracy, and Mean IoU, compared to the baseline DeiT model.
> >
> > **Q4. Visualizations on earlier layers and more heads of the transformers would help to strengthen your claim.**
> >
> > **Answer:** We appreciate the reviewer's suggestion on this matter. Accordingly, we have added the the visualizations from early to late layers as well as Figure 8 on alternative over-smoothing analysis on 2 datasets in Appendix D.2 of the revised document.
> >
> > -----
> > We hope we have cleared your concerns about our work. We have also revised our manuscript according to your comments, and we would appreciate it if we can get your further feedback at your earliest convenience.

---

> ### Author Response · Authors · 2024-11-22
> **Any Questions from Reviewer 7QSP on Our Rebuttal?**
>
> We would like to thank the reviewer again for your thoughtful reviews and valuable feedback.
>
> We would appreciate it if you could let us know if our responses have addressed your concerns and whether you still have any other questions about our rebuttal.
>
> We would be happy to do any follow-up discussion or address any additional comments.

---

### Official Review · Reviewer_CqxC · 2024-11-03

**Soundness:** 3
**Presentation:** 3
**Contribution:** 3
**Rating:** 8
**Confidence:** 3

**Summary:**

This paper propose the Twicing Attention, a novel attention mechanism that uses kernel twicing procedure in nonparametric regression to achieve slower decay of representational capacity and improved accuracy across various data modalities and tasks. And the design of this module builds on the study of the connection between self-attention and NLM smoothing filters. The method was tested on a public dataset, yielding promising results.

**Strengths:**

1. The paper is well-written, with clear expression of formulae and symbols, and will be highly readable.
2. The authors discuss the recurring problem of decay of representational capacity in Transformer, which has also been recognized as a manifestation of rank collapse in other studies. Instead of directly continuing the study of related work on rank collapse, the authors start with the NLM and try to gradually restore the cause of this phenomenon and again based on the proposed method that can alleviate this problem, the research angle is more interesting and also quite theoretical significance.
3. The author's description of the solution is complete and accompanied by thorough proofs, the process is clear and easy to understand, and the work done is very informative.

**Weaknesses:**

1. Admittedly, the authors' work is very rich and makes a very profound contribution at the theoretical level, but in my humble opinion, the authors' approach serves as a skillful level of reconciliation that moderates the rank collapse in depth, whereas a similar reconciliation skill is actually not uncommon in rank collapse-related research directions. I am not accusing the authors of not being innovative enough, but I hope that the authors can go further at the theoretical level and expand the sequence model that can really reduce the loss of information unlike the classical Transformer.
2. The author's research is more profound, but the experiments are less adequate, with too few test datasets and too few comparison methods. I tend to think that this is the result of too much time constraints, and I hope that the author will add more datasets as well as other experiments on the Transformer if there is enough time.

**Questions:**

1. For pure curiosity, I would like to ask what the authors think the performance of this method would be in more extreme cases, which in this case refers to two main scenarios: first, the performance on LLMs with a very large number of parameters. Second, on non-classical Transformer structures, such as Linear Transformer and other analogs.

---

> ### Author Response · Authors · 2024-11-20
> **Official Comment by Authors (1/2)**
>
> Thank you for your thoughtful review and valuable feedback. Below we address your concerns.
>
> -----
>
>
> **Q1. Admittedly, the authors' work is very rich and makes a very profound contribution at the theoretical level, but in my humble opinion, the authors' approach serves as a skillful level of reconciliation that moderates the rank collapse in depth, whereas a similar reconciliation skill is actually not uncommon in rank collapse-related research directions. I am not accusing the authors of not being innovative enough, but I hope that the authors can go further at the theoretical level and expand the sequence model that can really reduce the loss of information unlike the classical Transformer.**
>
> **Answer:** We thank the reviewer for endorsing the theoretical contributions of the paper. While we agree that manipulating spectrum in different ways is not utterly uncommon in literature, we believe that a modification--in a way that it further connects the attention mechanism to the well-established Twicing procedure in image processing [3] and nonparametric regression [4] with small bias property [1]--is a novel theoretical perspective. Unlike pure engineering heuristics or linear algebraic approaches to moderate the rank collapse, our work offers broader interpretability of Transformers along with the modified attention mechanism. Additionally, we believe that this interpretation can foster further research to study the potential benefits that such traditional statistical frameworks still has to offer for the modern deep learning theory.
>
> **Q2. The author's research is more profound, but the experiments are less adequate, with too few test datasets and too few comparison methods. I tend to think that this is the result of too much time constraints, and I hope that the author will add more datasets as well as other experiments on the Transformer if there is enough time.**
>
> **Answer:** We appreciate the reviewer's understanding of time constrains. We took our chance to carry out extra image segmentation experiments on another widely adopted dataset ADE20K and report the pixel accuracy, mean accuracy, and mean intersection over union (IoU) metrics to compare against the baseline in Table D below. We find that Twicing Attention offers improvements across all three metrics evaluated.
>
> **Table D:** Image segmentation on ADE20K.
> | Model | Pixel Acc. | Mean Acc. | Mean IoU |
> |-------|------------|-----------|----------|
> | DeiT  | 77.25      | 44.48     | 34.73    |
> | DeiT-Twicing | **77.51** | **45.53** | **35.12**
>
> Furthermore, we have done experiments with an additional competetitor model, NeuTRENO (Nguyen et al, 2023), that uses a nonlocal functional regularization to mitigate oversmoothing by constantly fusing with initial layer tokens. In the Table E below, we report the Top 1/Top 5 accuracy on ImageNet as well as their robustness against PGD, FGSM and SPSA adversarial attacks as in the paper. We observe that while both models outperform the baseline DeiT, our DeiT-Twicing offers relatively more improvements in almost all metrics.
>
> **Table E:** Image classification on ImageNet-1K.
> | Model | Top 1 | Top 5 | PGD Top1/Top5 | FGSM Top1/Top5 | SPSA Top1/Top5
> |-------|-------|-------|-----|------|---|
> | DeiT | 72.00 | 91.14 | 8.16 / 22.37 | 29.88 / 63.26 | 66.41 / 90.29
> | NeuTRENO | 72.44 | **91.40** | 8.85 / 23.83 | 31.43 / **65.96** | 66.98 / 90.48
> | DeiT-Twicing | **72.60** | 91.33 |**9.15** / **24.10**  | **32.28** / 65.67 | **67.12** / **90.53**

---

> ### Author Response · Authors · 2024-11-20
> **Official Comment by Authors (2/2)**
>
> **Q3. For pure curiosity, I would like to ask what the authors think the performance of this method would be in more extreme cases, which in this case refers to two main scenarios: first, the performance on LLMs with a very large number of parameters. Second, on non-classical Transformer structures, such as Linear Transformer and other analogs.**
>
> **Performance on LLMs:** To answer the question about the potential performance on LLMs with larger number of parameters, we trained a medium-sized model with 20.97M parameters compared to the small-model with 9.43M parameters. As a result, we observe that Transformer-Twicing still offers improvements across both validation and test perplexities indicating scaling properties about as good as the baseline Transformer. Also, in Figure 7 of the appendix, we show the training curves for the language models of both sizes, and observe that Twicing Attention helps the models converge relatively faster as well.
>
> **Table F:** Language modeling on Wikitext-103.
> | Model | Validation PPL ($\downarrow$) | Test PPL ($\downarrow$) |
> |-------|----------------|----------|
> | Transformer (small)| 38.11 | 37.51
> | +Twicing (small)| **37.12** | **36.69**
> | Transformer (medium)| 31.98 | 26.17
> | +Twicing (medium)| **30.91** | **25.65**
>
> **Extreme Unconventional Transformers:** Since the Twicing Attention's theoretical framework does not depend on how exactly the weight matrix is built, we believe that as long as any Transformer architecture-based model leverages an attention mechanism with a concrete similarity (attention) matrix that can be connected to either NLM denoising or nonparametric Nadaraya-Watson estimation as in the paper, Twicing is highly likely to offer extra representation capacity and robustness. In particular, as transformers with linear attention [9] are concerned, the implementation steps would involve denoting their separable similarity matrix in Eqn. (4) of [9] as $A$, and replacing it with the corresponding twicing weight matrix $2A-A^2$.
>
> **References:**
>
> [1]: Newey, W.K., F. Hsieh, and J.M. Robins (2004). "Twicing Kernels and a Small Bias Property of Semiparametric Estimators." Econometrica, Vol. 72, No. 3, pp. 947–962.
>
> [2]: Chernozhukov, V., Escanciano, J. C., Ichimura, H., Newey, W. K., & Robins, J. M. (2022). Locally robust semiparametric estimation. Econometrica: Journal of the Econometric Society.
>
> [3]: Tukey, J.W. (1977). "Exploratory Data Analysis". Reading, MA: Addison-Wesley.
>
> [4]: Stuetzle, W., and Y. Mittal (1979): "Some Comments on the Asymptotic Behavior of Robust Smoothers", in Smoothing Techniques for Curve Estimation, Lecture Notes, 757. New York: Springer-Verlag, 191–195.
>
> [9]: Katharopoulos, A., Vyas, A., Pappas, N., & Fleuret, F. (2020). Transformers are RNNs: Fast autoregressive transformers with linear attention. In Proceedings of the International Conference on Machine Learning (ICML). PMLR.
>
> -----
> We hope we have cleared your concerns about our work. We have also revised our manuscript according to your comments, and we would appreciate it if we can get your further feedback at your earliest convenience.

---

> > ### Author Response · Authors · 2024-11-22
> > **Additional results with unconventional attention mechanisms**
> >
> > We have conducted additional experiments with Linear Transformers [9] as described in our previous comment. Table A1 below compares the perplexities recorded for Linear Transformers with feature map $\phi(x) = \text{elu}(x)+1$ matching their original choice, and Linear-Twicing Transformers for which we apply the twicing transformation $2A-A^2$ where $A = \text{normalize}(\phi(Q)\phi(K)^\top)$. Note that we explicitly construct the similarity matrix $A$ for both of the models for our framework to work. On top of Table A1 results, we also observe relatively faster convergence for Linear-Twicing, very similar trend to what is illustrated in Figure 7 in the revised appendix of the paper. The positive results indicate that the applicability of Twicing Attention is not limited to standard softmax self-attention, but any reasonable similarity matrix can be covered. Lastly, we have added this results in Appendix B.1 and the appendix Table 6 of the revised document.
> >
> > **Table A1:** Validation/Test PPL on Wikitext-103 trained for 75 epochs.
> > | Model | Validation PPL | Test PPL |
> > |-------|----------------|----------|
> > | Linear Trans. | 40.00 | 41.26 |
> > | Linear-Twicing Trans. | **39.45** | **40.61**
> >
> > We hope this additional results complements our previous response to your question and clears your related concerns. We would be glad to hear your futher feedback on our work and rebuttal at your earliest convenience.

---

> ### Author Response · Authors · 2024-11-22
> **Any Questions from Reviewer CqxC on Our Rebuttal?**
>
> We would like to thank the reviewer again for your thoughtful reviews and valuable feedback.
>
> We would appreciate it if you could let us know if our responses have addressed your concerns and whether you still have any other questions about our rebuttal.
>
> We would be happy to do any follow-up discussion or address any additional comments.

---

> > ### Comment · Reviewer_CqxC · 2024-11-27
> > **Thanks to the author for thier detailed reply**
> >
> > Firstly, I'm very sorry for responding so late. The authors explained in detail my doubts about their method and added sufficient experiments to back it up (although I didn't ask for more experiments), the additional experiments added to my doubts and curiosity about the method, and I don't have any more questions to ask, I'm even willing to upgrade my rating because of such an informative response.

---

> > > ### Author Response · Authors · 2024-11-27
> > > **Thanks for your endorsement!**
> > >
> > > Thanks for your response, and we appreciate your endorsement.

---

> ### Author Response · Authors · 2024-11-24
> **Robustness of Twicing Attention within Unconventional Transformers**
>
> We would like to thank the reviewer again for your valuable initial reviews and feedback.
>
> We took our time to also test Linear-Twicing model's robustness againt Word Swap Attack using the same experimental setting as in Section 4.2 of the main text. As we show in Table B1 below, Linear-Twicing Transformer offers a significant improvement of almost 2 PPL points over the baseline Linear Transformer. This further validates that Twicing Attention can indeed enhance robustness of the model even with unconventional attention mechanisms. This result has been included in Table 7 of the revised document appendix.
>
> **Table B1:** Clean/Attacked Test PPL on Wikitext-103 under Word Swap attack.
> | Model | Clean PPL | Attacked PPL |
> |-------|----------------|----------|
> | Linear Trans. |  41.26 | 72.13
> | Linear-Twicing Trans. | **40.61** | **70.40**
>
> We hope this additional result provides additional justification of our insights provided in the previous replies and further addresses your question.

---

### Official Review · Reviewer_3adS · 2024-11-03

**Soundness:** 3
**Presentation:** 3
**Contribution:** 2
**Rating:** 6
**Confidence:** 3

**Summary:**

The self-attention mechanism's representational capacity diminishes significantly across layers, and this oversmoothing effect is reducing overall performance. This paper introduces Twicing Attention, a novel mechanism that connects self-attention computations with low-pass non-local means smoothing filters. By employing a kernel twicing procedure, it alleviates the low-pass effects of NLM smoothing while preserving meaningful information from residuals. Twicing Attention offers slower decay of representational capacity and improved accuracy across different data modalities. Significant performance improvement brought by Twicing attention is observed in multiple tasks.

**Strengths:**

1. Novelty: The authors introduce the Twicing Attention mechanism to slow down the eigenvalue decay associated with representational collapse.
2. Theoretical Contribution: the authors provide mathematical validation for the Twicing Attention’s capability to retain information across layers.
3. Experiments: the authors evaluate their proposed methods on both language models and vision models.

**Weaknesses:**

1. Limited improvement: The gains in clean data settings (such as on ImageNet in Tab. 1) are modest.
2. Lack of comparison: the work does not compare its method with alternative solutions that address oversmoothing, such as regularization strategies.
3. Lack of ablations: the authors are suggested to consider applying the proposed method at different layer depths or intervals and evaluate their difference.

**Questions:**

My question lies in the efficiency comparison (Tab. 4). Despite the fact that Twicing has the same complexity of $O(N^2 d)$ as claimed in the paper, it still increases the overhead by an additional 50% due to the extra matrix multiplication in line 7, Alg. 1. However, Tab. 4 indicates that implementing Twicing or not will not incur big difference on both speed and GFLOPs. What is the reason behind that? I would appreciate a more detailed efficiency analysis & comparison in the rebuttal phase if possible.

---

> ### Author Response · Authors · 2024-11-20
> **Official Comment by Authors (1/2)**
>
> Thank you for your thoughtful review and valuable feedback. Below we address your concerns.
>
> -----
>
>
> **Q1. Limited improvement: The gains in clean data settings (such as on ImageNet in Tab. 1) are modest.**
>
> **Answer:** We agree that Twicing Attention offers relatively modest accuracy improvements in clean data settings in Table 1. However, the clean data performance is not the main/only claim that we make about our model but improved overall accuracy (both under clean and corrupted data settings). Rather, we believe that the complimentary robustness comparisons make the Twicing model stand out as a substantially better model overall. In particular, Twicing models show capability to offer up to ~19\% improvement (FAN, PGD) with average of about ~8\% performance gains across all attacks. Besides, Figure 4 in the appendix shows that Twicing Attention can notably and consistently outperform the baseline across all 15 types of natural corruption types (about ~10% improvement on "contrast", "gaussian noise", and "impulse noise" to name but a few). Zooming out a little bit, this observation is interesting and consistent with the theory in the following sense: it suggests that Twicing models are inherently more stable across both clean and corrupted data settings by prioritizing stable representations over overtuned accuracy on clean accuracy—an aspect that can make models more susceptible to small perturbations, such as common adversarial attacks. Additionally, drawing on the small-bias property of Twicing kernels in nonparametric regression, one can argue that the resulting estimator is relatively less sensitive to bandwidth selection [1]. This reduced sensitivity mitigates the bias fluctuations often introduced by slight adjustments, making the estimator inherently more resilient to minor perturbations and improve model's robustness in general. Our experimental results under 3 widely adopted adversarial attacks validate that Twicing Attention is indeed significantly more robust compared to the baseline self-attention. We also refer to [2] for a more detailed robustness of twicing kernels in regression compared to the Nadaraya-Watson estimator kernel before twicing. At the same time, nonetheless, we also see in Table 4 of the revised document that improvements on clean and contaminated data for language modeling are comparable.
>
> [1]: Newey, W.K., F. Hsieh, and J.M. Robins (2004). "Twicing Kernels and a Small Bias Property of Semiparametric Estimators." Econometrica, Vol. 72, No. 3, pp. 947–962.
>
> [2]: Chernozhukov, V., Escanciano, J. C., Ichimura, H., Newey, W. K., & Robins, J. M. (2022). Locally robust semiparametric estimation. Econometrica: Journal of the Econometric Society.
>
> **Q2. Lack of comparison: the work does not compare its method with alternative solutions that address oversmoothing, such as regularization strategies.**
>
> **Answer:** We have conducted additional experiments comparing our method with an alternative model, NeuTRENO [8], that uses a nonlocal functional regularization to mitigate oversmoothing by constantly fusing with initial layer tokens. In the table below, we report the Top 1/Top 5 accuracy on ImageNet, as well as their robustness against PGD, FGSM and SPSA adversarial attacks. We observe that while both models outperform the baseline DeiT, DeiT-Twicing offers relatively larger improvements in almost all metrics.
>
> **Table C1:** ImageNet classification under clean and adversarially attacked settings.
> | Model | Top 1 | Top 5 | PGD Top1/Top5 | FGSM Top1/Top5 | SPSA Top1/Top5
> |-------|-------|-------|-----|------|---|
> | DeiT | 72.00 | 91.14 | 8.16 / 22.37 | 29.88 / 63.26 | 66.41 / 90.29
> | NeuTRENO | 72.44 | **91.40** | 8.85 / 23.83 | 31.43 / **65.96** | 66.98 / 90.48
> | DeiT-Twicing | **72.60** | 91.33 | **9.15** / **24.10**  | **32.28** / 65.67 | **67.12** / **90.53**
>
> [8]: Nguyen, T. M., Nguyen, T. M., & Baraniuk, R. (2023). Mitigating over-smoothing in transformers via regularized nonlocal functionals. NeurIPS 2023.

---

> > ### Comment · Reviewer_3adS · 2024-11-25
> > **Concerns mostly addressed**
> >
> > I would like to thank the authors for their rebuttal. My overhead concern is mostly addressed. I will raise my score.

---

> > > ### Author Response · Authors · 2024-11-25
> > > **Thanks for your consideration**
> > >
> > > Thank you for your response and support.
> > >
> > > We would greatly appreciate it if you could share any remaining concerns about our work so that we can address them before the rebuttal period concludes. We are more than happy to engage in follow-up discussions to resolve your concerns and kindly ask you to consider whether raising your score to 6 might better reflect your updated evaluation of our paper.
> > >
> > > Thank you once again for your time and thoughtful feedback!

---

> > > > ### Comment · Reviewer_3adS · 2024-11-25
> > > > **Thanks again for the rebuttal**
> > > >
> > > > Again, I sincerely thank the authors for their rebuttal.
> > > > The reason why I did not raise my score to 6 lies in some deep-rooted reason behind the topic (that might not be easy to rebuttal)
> > > > Firstly, I am not quite satisfied with the improvement. The amount of improvement, from my point of view, is not **that** significant in cases like clean data, Tab. 2 and 3. It is noteworthy that we are doing an extra multiplication that adds 1/2 of the attention calculation...
> > > >
> > > > Secondly, I am a bit of concerned about the actual applications of this method. In the day and age of LLMs, engineers to rely on off-the-shelf pretrained models very often. How could the proposed method be applied to off-the-shelf pretrained models? Could this be done with low training budget? I think this issue might need further clarification to enhance the applicability of this paper.
> > > >
> > > > Anyway, I do think this paper reaches the quality of ICLR from my knowledge and I won't object to the decision of acceptance despite an underrated score.

---

> ### Author Response · Authors · 2024-11-20
> **Official Comment by Authors (2/2)**
>
> **Q3. Lack of ablations: the authors are suggested to consider applying the proposed method at different layer depths or intervals and evaluate their difference.**
>
> **Answer:** In Appendix E, we compare 3 different choices of layer placements for Twicing Attention [1 to 12, 7 to 12, and 10 to 12]. As a result, we observe in particular that overall performance is roughly proportional to the number of Twicing layers in terms of clean data and under adversarial attacks. In Table C2 below, we report 2 more models--twicing at even layers, and using previous layer residual for twicing procedure for efficiency. We observe that even though DeiT-Twicing [even layers] has 6 Twicing Attention layers just as DeiT-Twicing [7-12], the latter model does better than the former. This validates that if one has capability to implement $n$ layers (out of  $L > n$ total layers) with Twicing Attention, it is better to place them as the latest n contiguous layers of the transformer.
>
>
> **Table C2:** Ablation of Twicing Attention placed at different layers.
> | Model              | Top 1 | Top 5 | Explanation |
> |--------------------|--------------------|--------------------|----|
> | DeiT               | 72.00              | 91.14              |
> | DeiT-Twicing [1-12]           | **72.60**              | 91.33              | Twicing Attention at all layers
> | DeiT-Twicing [7-12]           | 72.45              | **91.35**              | Twicing Attention at last 6 layers
> | DeiT-Twicing [10-12]           | 72.31              | 91.24              | Twicing Attention at last 3 layers
> | DeiT-Twicing [*even layers*]    | 72.42              | 91.28              | Twicing Attention at even layers
> | DeiT-Twicing [*overlayer residual*]  | 72.02              | 91.08             | Using previous layer residual
>
> **Q4. My question lies in the efficiency comparison (Tab. 4). Despite the fact that Twicing has the same complexity of as claimed in the paper, it still increases the overhead by an additional 50% due to the extra matrix multiplication in line 7, Alg. 1. However, Tab. 4 indicates that implementing Twicing or not will not incur big difference on both speed and GFLOPs. What is the reason behind that? I would appreciate a more detailed efficiency analysis & comparison in the rebuttal phase if possible.**
>
> **Answer:** We appreciate the reviewer’s attention to a potential source of confusion regarding Table 4. We elaborate on the details of that efficiency analysis as follows. Our model does not add 50% more computational cost as reported is the efficiency statistics considering the end-to-end flow of an input through the transformer. In fact, the additional computation--specifically calculating $A(V - A V)$ (with the pre-computed $AV$ as in Algorithm 1)--only marginally increases the total workload when considering the entire Transformer architecture. While this operation does add extra steps to the attention mechanism, the overall computational cost is dominated by other components, such as the feed-forward networks and linear transformations. These components combined require significantly more computation than the attention mechanism alone. Furthermore, attention layer itself is not doubled in terms of computational complexity since Twicing only adds an extra attention-weighted averaging while the basement of standard self-attention already consists of computing $QK^T$ and $\text{softmax}(\cdot)$ (which we do not repeat for Twicing) other than $AV$ matrix operation. As a result, theoretically, the added computation increases the total computational cost by only roughly about 7% (which is approximately consistent with Table 4 results) if we analyze a modestly simplified Transformer architecture in terms of component-wise runtime complexities and their contributions to the overall computational overhead. Considering the partial model, Twicing [10-12], we see that the additional overall computation is virtually negligible while offering a decent relative performance. It is also worth noting that since Twicing does not introduce any learnable parameters, its contribution to complexity of backward passes is minimal during pre-training.
>
> -----
> We hope we have cleared your concerns about our work. We have also revised our manuscript according to your comments, and we would appreciate it if we can get your further feedback at your earliest convenience.

---

> ### Author Response · Authors · 2024-11-22
> **Any Questions from Reviewer 3adS on Our Rebuttal?**
>
> We would like to thank the reviewer again for your thoughtful reviews and valuable feedback.
>
> We would appreciate it if you could let us know if our responses have addressed your concerns and whether you still have any other questions about our rebuttal.
>
> We would be happy to do any follow-up discussion or address any additional comments.

---

> ### Author Response · Authors · 2024-11-23
> **One more comparison with alternative model that address oversmoothing**
>
> On top of DeiT-NeuTRENO model compared in our previous response, we conducted an additional experiment with the FeatScale [10], another state-of-the-art vision transformer variant that tries to mitigate representation collapse. As shown in Table A2 below, our model DeiT-Twicing outperforms DeiT+FeatScale in both metrics on ImageNet classification. We report the same results in Table 9 of Appendix B.2 of our revision.
>
> **Table A2:** Top 1/Top 5 accuracies on clean ImageNet classification.
> | Model | Top 1 | Top 5 |
> |---|---|--
> | DeiT | 72.00 | 91.14 |
> | DeiT + FeatScale | 72.35 | 91.23 |
> | DeiT-Twicing | **72.60** | **91.33** |
>
> [10]: Peihao Wang, Wenqing Zheng, Tianlong Chen, and Zhangyang Wang. Anti-oversmoothing in deep vision transformers via the fourier domain analysis: From theory to practice. In International Conference on Learning Representations, 2022.
> ___
>
> We would appreciate it if you could let us know if our responses have addressed your concerns and whether you still have any other questions about our rebuttal. We would be happy to do any follow-up discussion or address any additional comments.
>
> If you agree that our responses to your reviews have addressed the concerns you listed, we kindly ask that you consider whether raising your score would more accurately reflect your updated evaluation of our paper. Thank you again for your time and thoughtful comments!

---

> ### Author Response · Authors · 2024-11-24
>
> We would like to thank the reviewer again for your valuable initial reviews and feedback.
>
> For additional robustness comparison, we benchmarked DeiT-NeuTRENO against ImageNet out-of-distribution and natural image corruptions, and found our model DeiT-Twicing being better in 3 out of 4 tests except ImageNet-A. However, NeuTRENO deteriorates the performance of the baseline DeiT in ImageNet-R, an observation which supports our model DeiT-Twicing to be **more stable** with *no extra hyper-parameters* introduced. This additional result has been included in Table 9 of the revised document appendix.
>
> | Model | ImageNet-A ($\uparrow$) | ImageNet-R ($\uparrow$) | ImageNet-C ($\downarrow$) | ImageNet-C (Extra)  ($\downarrow$) |
> |-------|------------|------------|------------|--------------------|
> | DeiT | 6.97 | 32.22 | 72.21 | 63.68 |
> | NeuTRENO | **8.36**  | 31.65 | 70.51 | 63.56
> | DeiT-Twicing [10-12] | *8.14* | *32.31* | **70.25** | *62.63*
> | DeiT-Twicing | 7.66 | **32.74** | *70.33* | **62.46** |
> ___
> We hope this additional results complement our previous response to your question and clears your related concerns. We would be glad to hear your futher feedback on our work and rebuttal at your earliest convenience.

---

> ### Author Response · Authors · 2024-11-28
> **Additional Results for Potential Use of Our Methods for LLM Finetuning**
>
> We sincerely appreciate the reviewer’s explanation of your score and your positive assessment of our paper's quality. We recognize that the applicability of any method to LLMs is a significant factor. While we agree with the reviewer that pretraining a large model with Twicing Attention applied at all layers may not be suitable for low-budget scenarios, we would like to share the following results and insights as a potential use case for Twicing Attention, both with off-the-shelf pretrained models and in full pretraining scenarios with lower budgets.
> ___
> **Fine-tuning a pretrained Switch Transformer.** To show how Twicing Attention can offer improvements to the pretrained models, we pretrain a medium sized (33M params) Switch Transformer [11], a Mixture of Experts architecture, with the standard self-attention on WikiText-103. Then we finetune this pretrained language model on Stanford Sentiment Treebank 2 (SST-2) dataset using standard self-attention (baseline) as well as Twicing Attention (ours) for 8 epochs. Table L1 compares Top 1 finetune test accuracies for both cases and we find that finetuning with Twicing Attention achieves higher accuracy, provided that the fine-tuning is long enough (usually a few more epochs than usual) for the model to adapt to the new attention mechanism.
>
>
> **Table L1:** Switch Transformer Pretrained on WikiText-103 and Finetuned on SST-2.
> | Mechanism | Fine-tune Test Acc. | #Params |
> |----|-----|----
> | Self-Attention | 77.78 | 33M
> | Twicing Attention | **78.34** | 33M
> ___
> **Partial Twicing model.** Additionally, we would like to highlight how DeiT-Twicing [10-12] (last 3 layers only) increases the FLOPs by **just over 1%** while improving robustness by 14.3% (ImageNet-A), 2.7% (ImageNet-C) [Table 2 of the paper] even surpassing the full model, and 5.5% (FGSM) [Table 1 of the paper]. We believe such a partially deployed Twicing Attention allows its application for almost negligible extra cost in practice.
> ___
> Thank you once again for your time and thoughtful feedback!
>
> **References:**
>
> [11]: Fedus, W., Zoph, B., & Shazeer, N. (2022). Switch transformers: Scaling to trillion parameter models with simple and efficient sparsity. Journal of Machine Learning Research.

---

> > ### Comment · Reviewer_3adS · 2024-11-30
> > **Future suggestions**
> >
> > I appreciate your effort in adding this experiment (score raised to 6), but I have a future request: could you please add some more analysis regarding finetuning on pretrained off-the-shelf large language models (e.g. LLaMA) in the future revision and conduct through evaluations using LLM eval benchmarks? I do understand that due to limited time in the discussion phase, you are unable to do this.

---

> > > ### Author Response · Authors · 2024-12-01
> > > **Thanks for your endorsement!**
> > >
> > > Thank you once again for your time and thoughtful feedback! We greatly appreciate your endorsement and suggestions regarding LLM fine-tuning. We will conduct the proposed experiments and incorporate additional analysis and evaluation of fine-tuning pre-trained off-the-shelf LLMs, such as LLaMA, using LLM evaluation benchmarks in our revision.

---

### Official Review · Reviewer_pR7y · 2024-11-04

**Soundness:** 3
**Presentation:** 3
**Contribution:** 2
**Rating:** 6
**Confidence:** 4

**Summary:**

The over-smoothing problem in Transformers is a well-known phenomenon, where the outputs of different attention layers in a Transformer model are highly similar. This paper introduces Twicing Attention to address this problem, which uses low-pass NLM smoothing filters to tackle this problem. The core idea can be phrased as, instead of using the standard attention matrix $A$, to use $2A - A^2$.

**Strengths:**

1. The paper is relatively easy to follow and well-written.
2. The proposed "Twicing Attention" is simple and easy to implement.
3. Theoretical motivation and the mathematical details behind their motivation and choices have been provided.

**Weaknesses:**

1. The paper compensates for the simplicity of the core idea by over-explaining and being overly verbose. For example, most of the material on pages 7-8 can be summarised in 2-3 paragraphs. Even Algorithm 1 on page 8 is redundant and too verbose. The algorithm's objective is clear and simple: to compute $2A - A^2$. I don't think one needs 12 lines to explain that.
2. Instead, the paper could have added to its contribution through a more thorough study. E.g., one avenue for improvement would be to consider other candidates besides the $2A - A^2$ and then compare them in the considered benchmarks

**Questions:**

I would be grateful if the authors could respond and address the weaknesses. I am willing to increase my score if the authors could address the weaknesses.

---

> ### Author Response · Authors · 2024-11-20
>
> Thank you for your thoughtful review and valuable feedback. Below we address your concerns.
>
> -----
>
>
> **Q1. The paper compensates for the simplicity of the core idea by over-explaining and being overly verbose. For example, most of the material on pages 7-8 can be summarised in 2-3 paragraphs. Even Algorithm 1 on page 8 is redundant and too verbose. The algorithm's objective is clear and simple: to compute $2A-A^2$. I don't think one needs 12 lines to explain that.**
>
> **Answer:** While we intended to make our narrative of NLM denoising and nonparametric regression perspective more comprehensible to the readers, we agree with the reviewer on the point that the content on the page 7, in particular, can be compressed into a more compact text. We have editted that section in our revision to achieve a concise alternative accordingly. We use the created space for extra insights on the robustness of Twicing Attention (Section 3.3), as well as extra experimental results (Section 4.1).
>
> **Q2. Instead, the paper could have added to its contribution through a more thorough study. E.g., one avenue for improvement would be to consider other candidates besides the $2A-A^2$ and then compare them in the considered benchmarks.**
>
> **Answer:** Even though we agree that there is still room for further empirical studies, we would argue that considering other "candidates" besides $2A-A^2$ is actually a little bit controversial since $2A-A^2$ is an only theoretically justified choice enabling us to study Twicing Attention through the lens of the well-established twicing procedure (which actually sets the paper title) in image reconstruction theory and nonparametric regression regime [3, 4]. The identity $(2A_\ell-A_\ell^2)V_\ell = A_\ell V_\ell + A_\ell(V_\ell - A_\ell V_\ell) = A_\ell V_\ell + A_\ell\cdot \text{r}_{\ell}$ is a quick way of reiterating the core motivation behind this very choice.
>
> For the sake of comparison and further insights, however, we have conducted additional experiments to study other candidates that are intended to approximate the twicing procedure without compromising the baseline efficiency. We report the results in Table A below. Note that each model in Table A is trained for 200 epochs. The compared models in this study are inspired by the core idea of twicing procedure--adding back the smoothed residual to the estimation. We observe that efficient approximations often exhibit faster initial convergence rates; however, they are less stable tend to fall behind the full model in later stages of training, as they struggle to capture and learn the more complex patterns which models are expected to learn in later stages. We still believe that such efficient versions can be made work well, yet we leave it for future research.
>
> **Table A:** Comparison of DeiT-Twicing and its efficient approximations as explained.
> | Model              | Top 1 | Top 5 | Explanation |
> |--------------------|--------------------|--------------------|----|
> | DeiT               | 66.85              | 88.06              |
> | DeiT-Twicing           | **67.43**             | **88.45**              |
> | Approx. Twicing [*overlayer residual*]  | 67.12              | 88.13              | Using previous layer residual $A_{\ell}(V_{\ell-1} - A_{\ell-1}V_{\ell-1})$ for twicing procedure for efficiency
> | Approx. Twicing [*temporal residual smoothing*]           | 67.08              | 88.06              | accumulating the residuals from previous layers with weight $\frac{\ell}{\ell+3}$ for $\text{r}_{\ell}$. This effectively smoothes the residuals temporally (without "spatial" smoothing via $A$)
> | Approx. Twicing [*local residual smoothing*] | 67.00 | 88.25 | Using $AV + \text{band}(A, w)(V-AV)$ where $\text{band}(A, w)$ extracts a banded part (diagonal strip) of $A$ of width $w \ll N$ for fast matrix multiplication.
>
> For further comparison with different candidates, we introduced a hyper-parameter into Twicing Attention as $AV + \lambda A(V-AV) = [(1+\lambda)A - \lambda A^2]V$. Then, we train this model on ImageNet classification with $\lambda = 1/2$, which lies right in the middle of baseline self-attention and our Twicing attention, so that it can capture the general effect of such scaling. We present our results in Table B below. While we find that it still offers improvements over the baseline, it falls behind the original Twicing Attention, and justifies the use of the proposed model with theoretical support.
>
> **Table B:** Comparison of DeiT-Twicing models using $2A-A^2$ and $(1+\lambda) A - \lambda A^2$ as a similarity matrix.
> | Model | Top 1 | Top 5 |
> |---|---|---|
> | DeiT | 72.00 | 91.14 |
> | Twicing ($2A-A^2$) | **72.60** | **91.33** |
> | Twicing ($(1+\lambda) A - \lambda A^2$) | 72.41 | 91.22
>
>
> -----
> We hope we have cleared your concerns about our work in this response and revised document. We would appreciate it if we can get your further feedback at your earliest convenience.

---

> > ### Comment · Reviewer_pR7y · 2024-11-23
> >
> > Thank you for your rebuttal and adding extra experiment for adding a hyperparameter to the twicing procedure. Your explanation partially addresses my questions and I am increasing my score to 6.

---

> > > ### Author Response · Authors · 2024-11-23
> > > **Thanks for your endorsement!**
> > >
> > > Thanks for your response, and we appreciate your endorsement.

---

> ### Author Response · Authors · 2024-11-22
> **Any Questions from Reviewer pR7y on Our Rebuttal?**
>
> We would like to thank the reviewer again for your thoughtful reviews and valuable feedback.
>
> We would appreciate it if you could let us know if our responses have addressed your concerns and whether you still have any other questions about our rebuttal.
>
> We would be happy to do any follow-up discussion or address any additional comments.

---

### Author Response · Authors · 2024-11-20
**Global Response**

Dear AC and reviewers,

First of all, we thank all reviewers for their endorsements as well as valuable feedback on our work. In particular, reviewers' positive comments on the clarity of our presentation (pR7y, CqxC, 7QSP), significance of our theoretical contribution (all 4 reviewers), and informativeness of the paper (CqxC, pR7y, 7QSP) have been encouraging for us.

In this global response, we would like to address some of the shared concerns among the reviewers as well as reiterate and clarify what major benefits Twicing Attention can offer, especially other than merely alleviating representation collapse.

We respond to each common concern as follows:

1. **Limited accuracy improvement on clean data.** We agree that Twicing Attention offers relatively modest accuracy improvements in clean data settings. However, the clean data performance is not the only claim that we make about our model but improved overall accuracy (both under clean and corrupted data settings). Rather, we believe that the complementary robustness comparisons make the Twicing model stand out as a substantially better model overall. In particular, Twicing models show capability to offer up to a significant ~19\% improvement (FAN, PGD) with average of about ~8\% performance gains across all adversarial attacks. Besides, Figure 4 in the appendix shows that Twicing Attention can notably and consistently outperform the baseline across all 15 types of natural corruption types (about ~10% improvement on "contrast", "gaussian noise", and "impulse noise" to name but a few). At the same time, nonetheless, we also see in Table 4 of the revised document that improvements on clean and contaminated data for language modeling are comparable.
2. **Additional computational cost for modest clean accuracy gain.** As mentioned in Point 1 above, the additional computation is also serving to obtain a significantly more robust model. In particular, notice how DeiT-Twicing is comparable to FAN against adversarial attacks while FAN introduces a more sophisticated architecture to achieve that. Additionally, refer to the relative improvements (\%) provided in Point 1. It is worth noting that most tailored robust models available also introduce similar (or sometimes more) computational complexity compared to Twicing (added a new paragraph in Related Works for this comparison). This is also sometimes known as the robustness-efficiency trade-off (RETO) which is hardly avoidable.
4. **Narrow problem formulation **(respresntation collapse)**.** While we genuinely understand why some reviewers tend to think that the paper only deals with representation collapse due to our problem introduction style, we would like to reiterate an almost equally important subject of our paper--improving the underlying theoretical denoiser/estimator framework through the twicing procedure [3, 4]--which also ensures more robustness [1, 2, 4] as it helps the model learn more stable representations. Furthermore, another importance of such a theoretical observation along with empirical justification is that it could foster interesting future research to explore more similar frameworks to improve deep learning models in various aspects. In light of this concern, we have adjusted our introduction and the following sections to give a little more importance to the robustness of Twicing Attention both theoretically and empirically.

### References:
[1]: Newey, W.K., F. Hsieh, and J.M. Robins (2004). "Twicing Kernels and a Small Bias Property of Semiparametric Estimators." Econometrica, Vol. 72, No. 3, pp. 947–962.

[2]: Chernozhukov, V., Escanciano, J. C., Ichimura, H., Newey, W. K., & Robins, J. M. (2022). Locally robust semiparametric estimation. Econometrica: Journal of the Econometric Society.

[3]: Tukey, J.W. (1977). "Exploratory Data Analysis". Reading, MA: Addison-Wesley.

[4]: Stuetzle, W., and Y. Mittal (1979): "Some Comments on the Asymptotic Behavior of Robust Smoothers", in Smoothing Techniques for Curve Estimation, Lecture Notes, 757. New York: Springer-Verlag, 191–195.

[5]: Victor Chernozhukov, Juan Carlos Escanciano, Hidehiko Ichimura, Whitney K. Newey, and James M. Robins (2022): "Locally robust semiparametric estimation". Econometrica, 90(4):1501–1535

[6]: Caron, M., Touvron, H., Misra, I., Jégou, H., Mairal, J., Bojanowski, P., & Joulin, A. (2021). Emerging properties in self-supervised vision transformers. Proceedings of the International Conference on Computer Vision (ICCV).

[7]: Darcet, T., Oquab, M., Mairal, J., & Bojanowski, P. (2024). Vision transformers need registers. Published as a conference paper at ICLR 2024.

-----

We are glad to answer any further questions you have on our submission.

---

### Author Response · Authors · 2024-11-20
**Summary of Revisions**

According to the comments and suggestions from reviewers, we have applied the following changes to the updated version of the paper:

1. General edit: We have cut the text on pages 7 and 8 to make the presentation more compact than before as suggested. To fill the created space, we have added more experiments, insights and theoretical explanations for the robustness of Twicing Attention, an aspect of our model which we seem not to have emphasized enough before. In particular, we associate the small bias property of the underlying twicing kernels to the reduction of bandwidth sensitivity [4, 1] and robustness to input perturbations through [5].
2. Extra experiments: We have conducted additional experiments on the image segmentation task on ADE20K dataset, and presented the comparison results between DeiT and DeiT-Twicing in Table 3 of the main text evaluated across three key metrics. We observe performance improvements over all metrics as reported. We have also provided the necessary experimental details in Appendix B.5. Besides, we have added a new model NeuTRENO (Nguyen et al, 2023) as a comparison model as requested. Also, we have trained a larger language modeling on Wikitext-103 to verify the scaling potential of Twicing Attention when implemented inside LLMs and obtained a positive answer as reported in Figure 7 and Table 6 of Appndix B.1.
3. Extra empirical analysis: As suggested by the reviewer 7QSP, we have provided the evolution of attention heatmaps for DeiT and DeiT-Twicing from early to late layers together with dozen of extra last layer heatmaps for more input images to strengthen our claims in Appendix D.2. We have also extended oversmoothing analysis in Figure 2 by conducting a similar experiment on ADE20K image segmentation task, and the results are positive and shown in Figure 8 in the appendix. In both cases, token smoothing is slower with Twicing Attention, validating our theoretical results.
4. Related works: We have added a discussion on the two papers [6, 7] studying the feature maps of Vision Transformers as suggested by the reviewer 7QSP since we found them indeed relevant. We have also added a new paragraph to the section dedicated to the research on robust transformer models building upon Point 1 of our Summary of Revisions.

### References:
[1]: Newey, W.K., F. Hsieh, and J.M. Robins (2004). "Twicing Kernels and a Small Bias Property of Semiparametric Estimators." Econometrica, Vol. 72, No. 3, pp. 947–962.

[4]: Stuetzle, W., and Y. Mittal (1979): "Some Comments on the Asymptotic Behavior of Robust Smoothers", in Smoothing Techniques for Curve Estimation, Lecture Notes, 757. New York: Springer-Verlag, 191–195.

[6]: Caron, M., Touvron, H., Misra, I., Jégou, H., Mairal, J., Bojanowski, P., & Joulin, A. (2021). Emerging properties in self-supervised vision transformers. Proceedings of the International Conference on Computer Vision (ICCV).

[7]: Darcet, T., Oquab, M., Mairal, J., & Bojanowski, P. (2024). Vision transformers need registers. Published as a conference paper at ICLR 2024.

---

### Author Response · Authors · 2024-11-22
**Additional Experimental Results**

Dear reviewers,

We would like to thank all reviewers again for your thoughtful reviews and feedback. We have obtained additional experimental result that validates the concept of Twicing Attention is not tied to the exact form of standard softmax self-attention, but it offers improvements for any reasonable similarity matrices including different types and forms of attention mechanisms as described below.

We have conducted additional experiments with Linear Transformers [9] as described in our previous comment. Table A1 below compares the perplexities recorded for Linear Transformers with feature map $\phi(x) = \text{elu}(x)+1$ matching their original choice, and Linear-Twicing Transformers for which we apply the twicing transformation $2A-A^2$ where $A = \text{normalize}(\phi(Q)\phi(K)^\top)$. Note that we explicitly construct the similarity matrix $A$ for both of the models for our framework to work. On top of Table A1 results, we also observe relatively faster convergence for Linear-Twicing, very similar trend to what is illustrated in Figure 7 in the revised appendix of the paper. The positive results indicate that the applicability of Twicing Attention is not limited to standard softmax self-attention, but it is compatible with any reasonable similarity matrix. We have appended this result to Appendix B.1 and Table 6 of the revised document (highlighted by blue color).

**Table A1:** Validation/Test PPL on Wikitext-103 trained for 75 epochs.
| Model | Validation PPL | Test PPL |
|-------|----------------|----------|
| Linear Trans. | 40.00 | 41.26 |
| Linear-Twicing Trans. | **39.45** | **40.61**

We would be happy to engage in any follow-up discussion or address any additional comments by the reviewers.

**References:**

[9]: Katharopoulos, A., Vyas, A., Pappas, N., & Fleuret, F. (2020). Transformers are RNNs: Fast autoregressive transformers with linear attention. In Proceedings of the International Conference on Machine Learning (ICML). PMLR.

---

### Meta-Review · Area_Chair_5BjA · 2024-12-22

**Metareview:**

This paper receives ratings of 8, 5, 6, 6, where the reviewers generally provided a positive assessment of this manuscript. In this paper, the authors propose Twicing Attention, a new self-attention mechanism designed to address representational collapse in transformers caused by over-smoothing. The authors established a connection between self-attention and nonparametric regression using twicing kernels. The proposed Twicing Attention mechanism leverages residuals between attention input and output, providing improvements in representational capacity and robustness. Experimental results also show taht Twicing Attention demonstrates consistent improvements across multiple tasks and benchmarks.

Strengths:
- The proposed Twicing Attention mechanism introduces a new perspective by leveraging residual information through twicing kernels, offering a novel solution to the representational collapse issue in transformers.
- Theoretical Rigor: this paper provides a strong theoretical foundation, connecting self-attention mechanisms to nonparametric regression techniques and demonstrating slower decay of representational capacity.
- Comprehensive experiments on vision (ImageNet, ADE20K) and language tasks (WikiText-103) showcase consistent performance improvements over baseline models. Enhanced robustness is also demonstrated under adversarial attacks and distribution shifts, indicating the practicality of the method.
- The mechanism is computationally efficient, requiring minimal additional overhead when selectively applied to specific layers of transformers.
- By addressing the limitation in transformer models (over-smoothing), the method has the potential to influence various applications in NLP, computer vision, and beyond.

Areas for Improvement:
- While the empirical results are compelling, comparisons with more diverse state-of-the-art methods could strengthen the claims o superiority and broader applicability.
- Some aspects of the theoretical framework, while rigorous, could be made more accessible to practitioners through intuitive explanations/visualizations.
- While the method is validated on several tasks, demonstrating its effectiveness rigorously across more diverse domains and larger datasets could enhance its impact.
- The paper briefly acknowledges limitations but could provide a more detailed discussion of areas for improvement and specific directions for future research.

The reviewers praised the paper’s strong theoretical underpinnings and empirical results, particularly its contributions to addressing over-smoothing in transformers. The paper also introduces a novel and elegant solution that aligns well with recent trends in enhancing transformer architectures. And therefore we recommend acceptance of this paper.

**Additional Comments On Reviewer Discussion:**

The authors and reviewers engaged in active and productive discussions during the rebuttal period, which helped refine and clarify the contributions of this manuscript. They highlighted its novel contributions and rigorous theoretical foundations. Minor concerns were raised regarding the clarity of some theoretical aspects and the scope of experimental comparisons, which were largely addressed in the authors’ rebuttal.

The authors addressed the reviewers’ concerns effectively, providing clarifications on theoretical aspects, additional ablation studies, and further discussions on the practical implications of their work. The authors also provided thoughtful responses about the method's limitations and its scalability to larger datasets and models, demonstrating awareness of future research directions.

---

### Decision · Program_Chairs · 2025-01-22

Accept (Poster)